# Bidirectional predictive coding

**Gaspard Oliviers**[1,2]**, Mufeng Tang**[1,2]**, & Rafal Bogacz**[1,2]
1. Nuffield Department of Clinical Neurosciences, University of Oxford, United Kingdom
2. MRC Centre of Research Excellence in Restorative Neural Dynamics, United Kingdom
Correspondence to: `rafal.bogacz@ndcn.ox.ac.uk`

## Abstract

Predictive coding (PC) is an influential computational model of visual learning and inference in the brain. Classical PC was proposed as a top-down generative model, where the brain actively predicts upcoming visual inputs, and inference minimises the prediction errors. Recent studies have also shown that PC can be formulated as a discriminative model, where sensory inputs predict neural activities in a feedforward manner. However, experimental evidence suggests that the brain employs both generative and discriminative inference, while unidirectional PC models show degraded performance in tasks requiring bidirectional processing. In this work, we propose bidirectional PC (bPC), a PC model that incorporates both generative and discriminative inference while maintaining a biologically plausible circuit implementation. We show that bPC matches or outperforms unidirectional models in their specialised generative or discriminative tasks, by developing an energy landscape that simultaneously suits both tasks. We also demonstrate bPC's superior performance in two biologically relevant tasks including multimodal learning and inference with missing information, suggesting that bPC resembles biological visual inference more closely.

## 1 Introduction

Visual inference plays a critical role in the brain, providing information processing for interpreting and interacting with the environment. Two frameworks have emerged to explain how the brain could implement visual inference. The first describes vision as a bottom-up discriminative process, where sensory stimuli are progressively filtered through layered neural architectures, predicting behaviourally relevant outputs (Hubel & Wiesel, 1962). This framework resembles inference in feedforward neural networks commonly employed in machine learning for image classification. The other framework formalises vision as a generative process, where the brain constructs a probabilistic model of sensory inputs (Knill & Pouget, 2004). From this perspective, the brain learns a top-down generative model with priors over incoming sensory activity, and neural responses arise from the Bayesian inversion of this model, estimating posterior probabilities of brain states given sensory information. Various neural implementations of this inversion have been proposed, including variational methods (Friston, 2005; 2010) or sampling approaches (Fiser et al., 2010; Orbán et al., 2016; Haefner et al., 2016). Experimental evidence suggests that visual perception may be a combination of both frameworks (Teufel & Fletcher, 2020; Peters et al., 2024). Discriminative models explain the rapid initial responses to visual stimuli through efficient bottom-up processing of the brain (Peters et al., 2024), whereas generative models capture the probabilistic computations critical for optimally integrating noisy sensory inputs with prior knowledge, as displayed in perception and behaviour (Ernst & Banks, 2002; Wolpert et al., 1995; Knill & Richards, 1996; van Beers et al., 1999).

One computational model capable of capturing both generative and discriminative processing modes in the brain is predictive coding (PC). In its generative formulation, PC accounts for a broad range of biological data from the visual system (Srinivasan et al., 1982; Rao & Ballard, 1999; Hosoya et al., 2005) and is successful in tasks such as associative memory (Salvatori et al., 2021) and image generation (Oliviers et al., 2024). In its discriminative formulation, PC matches backpropagation-trained neural networks in image classification (Whittington & Bogacz, 2017; Pinchetti et al., 2025) while also explaining neural data (Song et al., 2024). Crucially, PC relies on local computations

and Hebbian learning rules consistent with biological constraints (Hebb, 1949; Posner et al., 1988; Lisman, 2017) and implementable in realistic neural circuits (Bogacz, 2017), making it a strong candidate for modelling both generative and discriminative learning in the brain. However, current PC formulations remain restricted to a single inference mode. Hybrid approaches that combine generative and discriminative versions exist (Tscshantz et al., 2023) but sacrifice performance in at least one domain, leaving open how a single PC model could flexibly support both inference modes as required for biological vision.

In this work, we propose bidirectional predictive coding (bPC), a novel model of biological vision. bPC provides a biologically grounded neural mechanism that explains how the brain can simultaneously perform generative and discriminative inference based on PC. We focus on evaluating and understanding bPC's performance in a range of computational tasks, where PC models with either generative or discriminative inference have been individually successful. Our key contributions can be summarised as follows:

- We propose bidirectional predictive coding, a biologically plausible model of visual perception employing both generative and discriminative inference that naturally arises from minimising a single energy function;

- We show that bPC performs as well as its purely discriminative or generative counterparts on both supervised classification tasks and unsupervised representation learning tasks, and outperforms precedent hybrid models;

- We provide an explanation for the superior performance of bPC in both tasks, by showing that it learns an energy landscape that better captures the training data distribution than its unidirectional counterparts;

- We further show that bPC outperforms other PC models in two biologically relevant tasks, including learning in a bimodal model architecture and inference with occluded visual scenes, indicating its potential as a more faithful model for visual inference in the brain.

## 2 BACKGROUND AND RELATED WORK

**Discriminative predictive coding.**   Recent research has explored PC models that employ bottom-up predictions from sensory inputs to latent neural activities (Whittington & Bogacz, 2017; Song et al., 2024), analogous to traditional feedforward neural networks. These models are structured as hierarchical Gaussian models consisting of $L$ layers, each characterized by neural activity $x_l$, where $x_1$ corresponds to the sensory input and $x_L$ represents a label. PC learns this hierarchical Gaussian structure by minimizing an energy function corresponding to the negative joint log-likelihood of the model:

$$E_{disc}(x, V) = \sum_{l=2}^{L} \frac{1}{2}\|x_l - V_{l-1}f(x_{l-1})\|_2^2, \qquad (1)$$

where $x$ denotes the set of neural states from $x_1$ to $x_L$, the parameters $V$ include the bottom-up weights $V_l$ of each layer $l$, and $f$ represents an activation function. We refer to this model as discriminative predictive coding (discPC) and illustrate it in the top panel of Figure 1A. In discPC, inference is performed by a forward pass from $x_1$ to $x_l$'s, and the learning rule of this model approximates BP, using only local computations and Hebbian plasticity (Whittington & Bogacz, 2017). Recent work showed that discPC performs comparably to BP in classifying MNIST, Fashion-MNIST and CIFAR-10 (Pinchetti et al., 2025) and outperforms BP at learning scenarios encountered by the brain, such as online and continual learning (Song et al., 2024). However, discPC lacks unsupervised learning capabilities due to the non-uniqueness of the solution to its generative dynamics (Sun & Orchard, 2020).

**Generative predictive coding.**   Classically, PC uses top-down predictions from neural activities to sensory data, based on the hypothesis that the brain learns by minimizing the error between its predicted sensory inputs and the actual sensory input (Rao & Ballard, 1999; Friston, 2005). With neurons arranged hierarchically, the negative joint log-likelihood can be written as:

$$E_{gen}(x, W) = \sum_{l=1}^{L-1} \frac{1}{2}\|x_l - W_{l+1}f(x_{l+1})\|_2^2, \qquad (2)$$

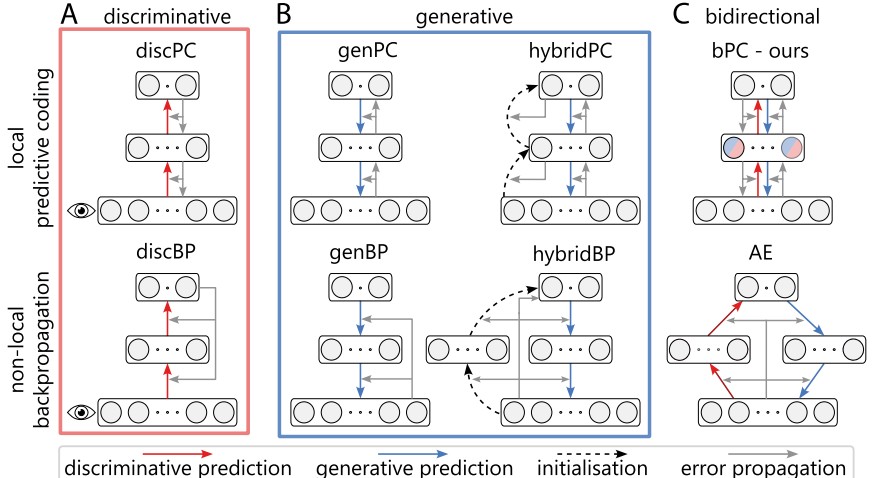

Figure 1: **Model architectures used in our experiments.** Arrows indicate the direction of prediction and error propagation. Dashed arrows represent initialisation. Discriminative models (red) are parametrised by bottom-up mappings from sensory inputs to brain states, generative models (blue) are parametrised by top-down mappings from brain states to sensory inputs, and bidirectional models incorporate both directions. Models in the top row employ local computations and error propagation that are considered biologically plausible, whereas those in the bottom row utilise non-local, backpropagation-based error computations that lack biological plausibility.

where $x_1$ is set to sensory data and $x_L$ can be clamped to a label for supervised learning or remain free for unsupervised learning. The parameters $W_l$ are top-down weights. We refer to this formulation of predictive coding as generative predictive coding (genPC). In genPC, inference of latent activities $x_l$ is performed by clamping $x_1$ to sensory inputs and iteratively updating $x_l$'s via gradient descent on the energy $E_{gen}$. genPC's top-down predictions are illustrated in the top-left panel of Figure 1B. genPC has explained various visual phenomena, such as extra-classical receptive field effects and repetition suppression (Rao & Ballard, 1999; Hohwy et al., 2008; Auksztulewicz & Friston, 2016). More recently, genPC has been employed to model associative memory (Salvatori et al., 2021) and unsupervised image generation (Oliviers et al., 2024; Zahid et al., 2024). The computational framework of genPC can be implemented within neural networks that utilize local computations and Hebbian plasticity (Bogacz, 2017). However, genPC has poor performance in supervised learning tasks (Tscshantz et al., 2023).

**Predictive coding models with mixed inference modes.** In this work, we benchmark our bPC model primarily against hybrid predictive coding (hybridPC) (Tscshantz et al., 2023). In hybridPC, iterative inference proceeds in the same way as in genPC. However, an additional bottom-up network is introduced to provide a feedforward initialisation of neural activities. This network only sets the initial states and does not influence the subsequent dynamics, as illustrated in the top-right panel of Figure 1B. Although hybridPC performs both supervised and unsupervised learning, its supervised performance falls short of discPC. In this work, we show that bPC performs supervised learning on par with discPC, and we provide an explanation of hybridPC's inferior performance through the lens of bPC (see SM H). Sun & Orchard (2020) noted the energy-minimising nature of PC models could theoretically allow image generation in discPC (Eq. 1), by clamping $x_L$ to a label and iteratively updating $x_1$. However, the generated images appear nonsensical due to the non-unique inferential dynamics solutions. Salvatori et al. (2022) generalised the idea of Sun & Orchard (2020) to a PC model where all neurons are interconnected with each other. However, it only slightly outperforms the classification performance of a linear classifier. Finally, Qiu et al. (2023) proposed a bidirectional PC model in which the connections between separate layers share the same weights, e.g. the bottom-up weights from layer $l-1$ to $l$ equal the top-down weights from layer $l$ to $l+1$. It is unlikely that the brain shares synaptic connections between separate layers of processing.

**Discriminative and generative models of the brain.** Cortical processing models are often divided into discriminative, which use feedforward networks to filter inputs (Yamins et al., 2014; Fukushima, 1980), and generative, grounded in the Bayesian Brain hypothesis (Helmholtz, 1866; Knill & Pouget, 2004) and theories such as predictive coding (Rao & Ballard, 1999; Friston, 2005) or adaptive resonance (Grossberg, 2012). Growing evidence suggests the brain combines these ap-

proaches (Lamme & Roelfsema, 2000; Teufel & Fletcher, 2020; Peters et al., 2024): for example, it might employ a discriminative approach to rapidly extract important features, and perform Bayesian inference on these features to form high-level perceptions such as object categories (Yildirim et al., 2024). Few models explicitly integrate both. The wake-sleep algorithm used to train the Helmholtz machine (Dayan et al., 1995) is analogous to a generative-discriminative process. The Symmetric Predictive Estimator (Xu et al., 2017) models bidirectional processing, though only on toy tasks. A related bidirectional model based on information maximization (Bozkurt et al., 2023) improves biological plausibility but requires two inference phases per update. Interactive activation models (Rumelhart & McClelland, 1982) also perform bidirectional inference, but do not incorporate a learning mechanism.

**Discriminative and generative models in machine learning.** Research in machine learning has likewise sought to integrate discriminative and generative pathways. Encoder–decoder architectures such as U-Nets (Ronneberger et al., 2015) combine bottom-up and top-down processing via skip connections, while models including VAVAE (Yang & Wang, 2025) and VAR (Tian et al., 2024) unify both pathways within a shared latent representation. Other approaches, exemplified by PredNet (Lotter et al., 2016), take more explicit inspiration from cortical circuitry to implement bidirectional information flow. Despite their strong empirical performance, these models depend on non-local learning signals and therefore do not provide a biologically plausible account of bidirectional learning in the brain.

## 3 BIDIRECTIONAL PREDICTIVE CODING

In contrast to genPC and discPC, bPC neurons perform both top-down and bottom-up predictions, as shown in the top panel of Figure 1C. bPC achieves this bidirectional inference by unifying the energy functions of genPC and discPC into a single formulation, enabling both generative and discriminative prediction within the same circuit. Using the notations previously introduced, the energy function of bPC given by:

$$E(x, W, V) = \sum_{l=1}^{L-1} \frac{\alpha_{gen}}{2} \|x_l - W_{l+1} f(x_{l+1})\|_2^2 + \sum_{l=2}^{L} \frac{\alpha_{disc}}{2} \|x_l - V_{l-1} f(x_{l-1})\|_2^2, \quad (3)$$

where $W_l$ are the top-down weights and $V_l$ are the bottom-up weights. $\alpha_{gen}$ and $\alpha_{disc}$ are scalar weighting constants, which are needed to account for magnitude differences in the errors of bottom-up and top-down predictions. The weighting constants can be viewed as learnable precision parameters (Friston, 2005); however, they are tuned and kept constant in our implementation for simplicity.

In each trial of learning, we first initialise the layers of neural activity using a feedforward sweep from layer $x_1$ to $x_L$ along the bottom-up predictions, similar to discPC (Whittington & Bogacz, 2017). For instance, the second and third layers are initialised as $x_2 = V_1 f(x_1)$ and $x_3 = V_2 f(V_1 f(x_1))$ respectively. This initialisation strategy can be interpreted as a mechanism for fast amortised inference when a sensory input is initially encountered, which is also observed in the brain (Thorpe et al., 1996; Lamme & Roelfsema, 2000). All bPC experiments used this activity initialisation scheme. After, we update the neural activities to minimise $E$ via several gradient descent steps (neural dynamics) following:

$$\frac{dx_l}{dt} \propto -\nabla_x E = -\epsilon_l^{gen} - \epsilon_l^{disc} + f'(x_l) \odot \left( W_l^\top \epsilon_{l-1}^{gen} + V_l^\top \epsilon_{l+1}^{disc} \right) + \mathcal{N}(0, \sigma^2 I), \quad (4)$$

where

$$\epsilon_l^{gen} := \alpha_{gen}(x_l - W_{l+1} f(x_{l+1})), \quad \epsilon_l^{disc} := \alpha_{disc}(x_l - V_{l-1} f(x_{l-1})) \quad (5)$$

denote the top-down and bottom-up prediction errors of neurons in layer $l$ respectively. $f'$ denotes the derivative of the function $f$, and $\odot$ is the element-wise product. The normally distributed noise $\mathcal{N}$ is zero-mean, temporally uncorrelated, and independent across neurons. By default, we set $\sigma^2 = 0$, yielding deterministic dynamics that converge to the maximum a posteriori estimate. Unless stated otherwise, all experiments in this paper use these inference dynamics. Setting $\sigma^2 = 1$ induces stochastic dynamics that sample from the model's posterior (Oliviers et al., 2024). These dynamics enable learning the distributions of sensory inputs.

After updating neural activities, the weights are updated to minimise $E$ via a single gradient descent step:

$$\Delta W_l \propto -\nabla_{W_l} E = \epsilon_{l-1}^{gen} f(x_l)^\top, \quad \Delta V_l \propto -\nabla_{V_l} E = \epsilon_{l+1}^{disc} f(x_l)^\top. \quad (6)$$

## 3.1 NEURAL IMPLEMENTATION

The computations described in Eqs 4 and 6 can be implemented in a neural network with fully local computation and plasticity, as illustrated in Figure 2. This network contains value neurons, which encode $x_l$, error neurons, which represent prediction errors, and synaptic connections, which encode the model parameters. All computations are local, relying solely on pre- and post- synaptic activity. Value neuron dynamics depend on local error signals, their own activity, and incoming synaptic weights. Similarly, error neuron activity depends only on adjacent value neurons and synaptic weights. The plasticity of the weights is also Hebbian, as it is a product of pre- and post-synaptic activity. The local implementation of bPC inherits that of genPC or discPC (shown in Figure 2 right), although it has two distinct error neurons per value neuron for bottom-up and top-down prediction errors.

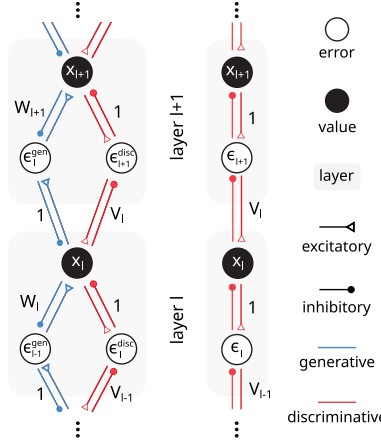

Figure 2: Neural implementation of bPC (left) and discPC (right).

## 3.2 FLEXIBLE LEARNING

bPC can be trained both in supervised and unsupervised settings. In all cases, neurons in intermediate layers (layer 2 to $L-1$) are un-clamped and evolve according to the neural dynamics in equation 4. In supervised settings, the first layer $x_1$ is clamped to the input, while the top layer $x_L$ is clamped to the target labels. In unsupervised settings, $x_L$ is left unclamped, allowing bPC to learn compressed representations of the input. A mixed setting is also possible, where only a subset of neurons in $x_L$ are clamped to label information, while others remain free. In this setting, the model can jointly infer labels and learn an compressed representation.

# 4 EXPERIMENTS

## 4.1 BPC PERFORMS SIMULTANEOUSLY WELL IN CLASSIFICATION AND GENERATION

In this experiment, we assessed bPC's capacity for both classification (discriminative) and class-conditional image generation (generative). We compared bPC with discPC, genPC, hybridPC, and their backpropagation equivalents on MNIST and Fashion-MNIST, using identical architectures with two hidden layers of 256 neurons each. Additional baselines, including bPC with shared bottom-up and top-down weight as proposed by Qui et al (2023), are provided in the supplementary material (SM) C.

During training, the input layer $x_1$ was clamped to images and the output layer $x_L = x_4$ to their corresponding one-hot labels (Figure 3A). After training, discriminative performance was assessed by fixing $x_1$ to an input image and inferring the label at $x_L$ over 100 inference steps. Generative performance was evaluated by clamping $x_L$ to a class label and measuring the root mean squared error

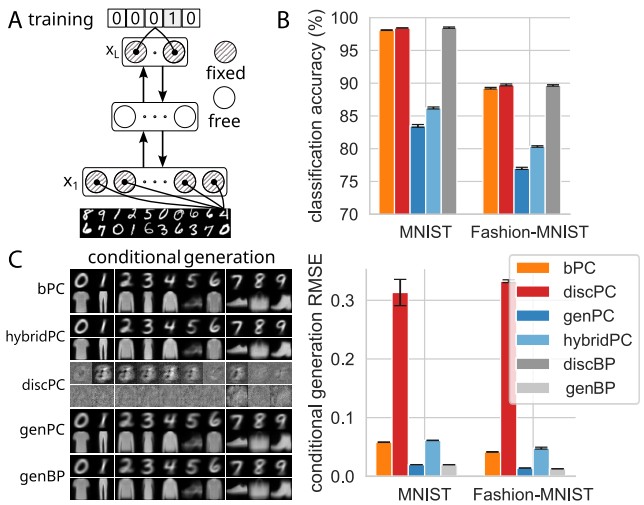

Figure 3: **bPC accurately classifies and generates class average images on MNIST and Fashion-MNIST.** A: Training set-up. The models are trained with $x_1$ fixed to images and $x_L$ fixed to labels. B: Classification accuracy of models. C: Examples of the generated images conditional on class labels (left) and RMSEs between generated images and mean images of each class (right). Error bars denote the standard error of the mean (s.e.m.) across 5 seeds.

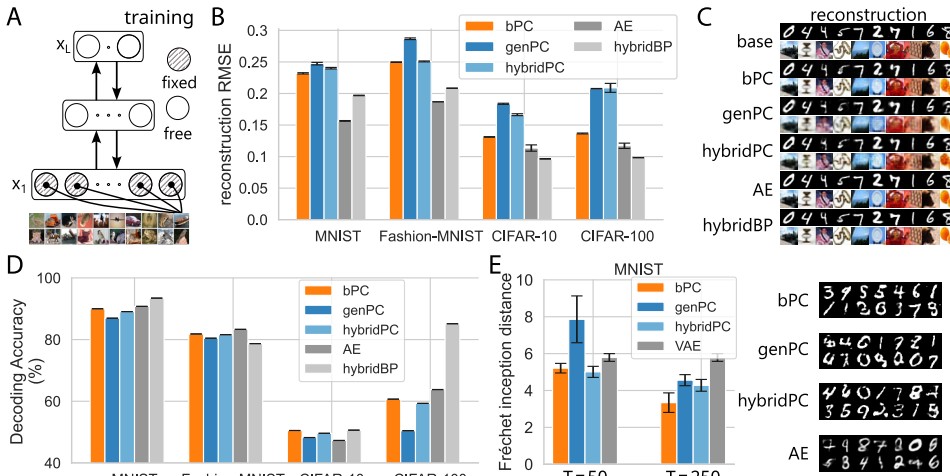

Figure 4: **bPC matches the performance of hybridPC at unsupervised learning.** A: Training set-up, where only $x_1$ is clamped to input images. B: Reconstruction RMSE from representations. C: Example reconstructions. D: Linear decoding accuracy from representations. E: Fréchet Inception Distance of samples generated by models with stochastic dynamics, with examples shown. T denotes the number of inference steps used during training. Error bars show s.e.m. across 5 seeds.

(RMSE) between the inferred image at $x_1$ (after 100 inference steps) and the average image for that class.

The results are shown in Figure 3B. On both datasets, bPC achieved classification accuracy comparable to discPC and discBP, while genBP and hybridPC performed less well. For generation, bPC obtained RMSE scores similar to genPC, hybridPC, and genBP, whereas discPC exhibited much higher errors. Visualizations in Figure 3C further illustrate these differences: discPC generated images with little class-relevant structure, while bPC and other bidirectional models produced clear and representative samples.

These findings align with previous reports that unidirectional PC models excel only in their specialized domain (Sun & Orchard, 2020; Tscshantz et al., 2023), and demonstrate that bPC can integrate both discriminative and generative capabilities within a single framework. Notably, this integration uses the same number of error neurons but only half the value neurons required for maintaining separate unidirectional pathways for the two tasks. This indicates that bPC is more energy-efficient, especially as error signals can be handled in the dendrites of neurons rather than by a separate neuron population (Mikulasch et al., 2023). We observed that the discriminative weighting parameter ($\alpha_{disc}$) must be set higher than the generative weighting ($\alpha_{gen}$) due to the larger magnitude of top-down prediction errors. An exploration of the trade-off between $\alpha_{disc}$ and $\alpha_{gen}$ is provided in SM D.

## 4.2 bPC PERFORMS UNSUPERVISED REPRESENTATION AND DISTRIBUTION LEARNING

Next, we show that bPC learns compressed representations and data distributions in the absence of supervision. We compared bPC, genPC, hybridPC, and their backpropagation (BP) equivalents on MNIST, Fashion-MNIST, CIFAR-10, and CIFAR-100, excluding discPC since it cannot perform unsupervised learning. We clamped only $x_1$ to input images, leaving all other layers free (see Figure 4A). For MNIST and Fashion-MNIST, models used two hidden layers (256 neurons each) and a 30-neuron representation layer $x_L$, trained with only 8 inference steps per update to test fast inference. CIFAR models had five convolutional layers and a 256-neuron representation layer, trained with 32 inference steps. An activity decay term at $x_L$ stabilized learning and regularized representations. After training, representations were obtained from $x_L$ using 100 inference steps. We evaluated these representations in two ways. First, reconstruction quality was measured by reinitializing layers $x_1$-$x_{L-1}$, clamping $x_L$, and running 100 inference steps, with RMSE computed against the input. Second, linear readout classification accuracy was measured from $x_L$.

To assess distribution learning, we compared bPC, genPC, hybridPC, and a VAE (Kingma et al., 2013) on MNIST, extending all PC models with stochastic dynamics to infer full posteriors ($\sigma^2 = 1$

in equation 4, see SM A.5 for more details). Architectures matched those above, trained with 50 or 250 inference steps. Models were evaluated using Fréchet Inception Distance (FID). Samples were generated ancestrally by following top-down predictions starting from sampling $x_L$ from its Gaussian prior and propagating downward through conditional Gaussians.

As shown in Figure 4B-E, bPC consistently outperformed genPC, yielding lower reconstruction RMSE, higher decoding accuracy, and better FID. It matched hybridPC across most tasks, and on CIFAR datasets significantly surpassed it in reconstruction error, approaching BP-based baselines. Reconstructions and generated samples (Figures 4C, E) illustrate these differences.

These results highlight that bPC outperforms genPC when inference steps are limited, and matches hybridPC despite the latter's amortized pathway. This suggests that bPC's bottom-up pathway also serves as amortized inference, rapidly initializing neural activities toward the optimal state. The slight edge of bPC over hybridPC likely arises from the active involvement of bottom-up weights $V_l$ during iterative inference, which continuously deliver sensory information to latent neurons. This effect becomes more pronounced with complex inputs such as CIFAR.

### 4.3 bPC PERFORMS COMBINED SUPERVISED AND UNSUPERVISED LEARNING

In this experiment, we combined the supervised and unsupervised settings described above to test whether bPC can simultaneously develop discriminative capabilities and compact representations. This setting is motivated both computationally and biologically: real-world learning rarely occurs in isolation, and cortical circuits appear to integrate categorical supervision (e.g., from higher-order areas) with unsupervised structure learning from sensory input.

We trained bPC, hybridPC, and their BP equivalents on MNIST, Fashion-MNIST, and CIFAR-10. During training, the input layer $x_1$ was clamped to images, while the top layer $x_L$ was partially clamped to one-hot labels, leaving the remaining neurons free to learn complementary representations (Figure 5A). For MNIST and Fashion-MNIST, models had two hidden layers of 256 neurons each, and $x_L$ comprised 40 neurons (10 for labels, 30 for representations). For CIFAR-10, we used a convolutional architecture with four hidden convolutional layers and a 266-neuron latent layer ($x_L$, with 10 label and 256 representation neurons). Activity decay was applied to the representational subpopulation in $x_L$ to regularize learning. Classification accuracy was evaluated as in Section 4.1, by presenting only images and inferring labels at $x_L$. Generative quality was assessed as in Section 4.2, but with reconstructions conditioned jointly on the inferred representation and the label.

Figure 5B reveals that bPC achieves classification accuracy on par with BP models across all datasets. Furthermore, it significantly surpasses hybridPC, particularly on CIFAR-10, where hybridPC exhibits more than 45% lower accuracy than bPC. In terms of generative quality, Figure 5D shows that bPC achieves reconstruction errors similar to hybridPC across all datasets. The reconstructed examples in Figure 5C illustrate that bPC goes beyond generating class-average images, capturing features such as shape, color, and spatial locations (e.g., it generates '4's in different styles). However, fine details are absent in bPC's CIFAR-10 reconstructions due to artefacts introduced by max-pooling operations in the discriminative bottom-up pathway, affecting the

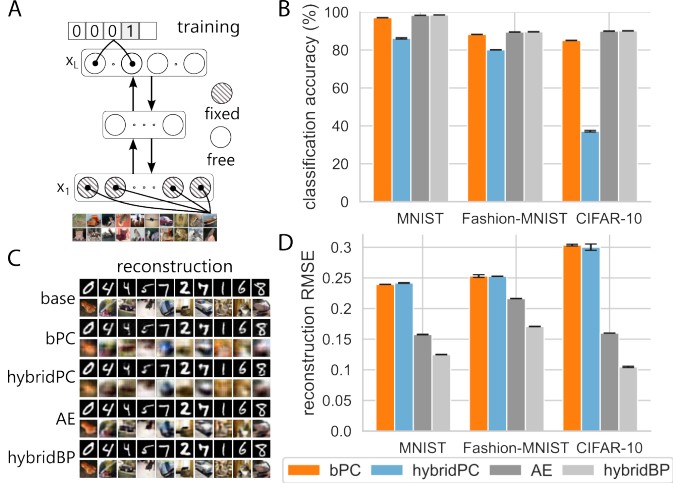

Figure 5: **bPC is the only PC model that can jointly learn low-dimensional representations of images and accurately classify them.** A: Training set-up, where the latent layer is only partially clamped to class labels. B: Classification accuracy. C: Example reconstructions on MNIST and CIFAR10. D: Reconstruction RMSEs. Error bars show s.e.m. across 5 seeds.

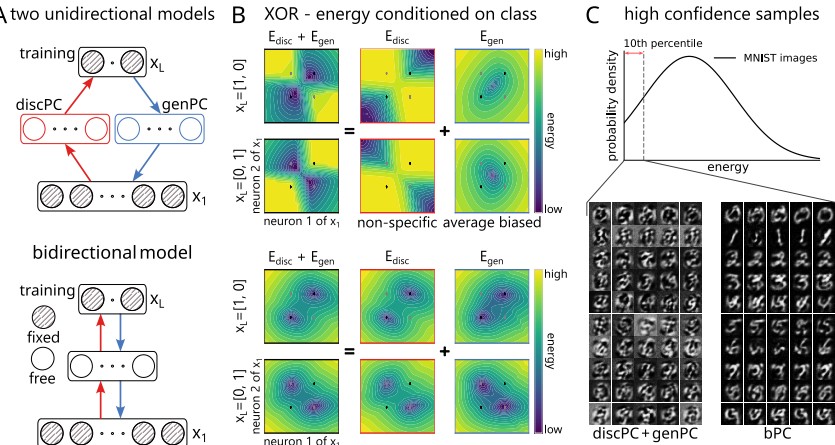

Figure 6: **bPC develops energy landscape suitable for both generation and classification.** A: We compare bPC (bottom) to training a discPC and a genPC separately (top). B: Visualisation of the models' energy landscape after training on XOR. Shown are the discriminative, generative, and summed energies. C: MNIST samples considered highly likely (bottom 10% energy) by the models.

generative top-down reconstruction process. Consequently, bPC shows stronger grid-like artefacts than hybridPC, which uses only max-pooling for initialization, and than BP models with separate generative and discriminative layers. While removing max-pooling removes artefacts, it reduces classification accuracy (see SM E). Nonetheless, even without max-pooling, bPC's classification accuracy significantly outperforms hybridPC, while generative performance remains comparable. Future research could refine bPC's architecture to balance classification accuracy with artefact-free generation.

### 4.4 BPC'S SHARED LATENT LAYERS PREVENT BIASED OR OVERCONFIDENT ENERGY LANDSCAPE

In this experiment, we investigated why bPC can perform both discriminative and generative tasks effectively, focusing on the role of its shared latent layers in shaping the energy landscape (Eq. 3). As a toy example, we considered the XOR problem, where the landscape can be directly visualized. We trained bPC, discPC, and genPC (each with two hidden layers of 16 neurons) by clamping $x_1$ to 2D inputs of XOR and $x_4$ to scalar outputs (Figure 6A). bPC used the same number of parameters as discPC and genPC combined but half the neurons.

An ideal XOR landscape has well-localized minima only at valid input-label pairs ([0,1] and [1,0] for one class; [0,0] and [1,1] for the other). Figure 6B top shows that discPC instead develops broad low-energy regions, indicating overconfidence even for implausible or out-of-distribution (OOD) inputs. genPC collapses each class into a single mean, failing to capture the true structure. A combined discPC+genPC model inherits both flaws. In contrast, bPC learns sharp, class-specific minima centred on valid inputs (Figure 6B bottom).

To test generality, we evaluated the models from Section 4.1 by sampling MNIST digits. We clamped $x_L$ to a label, initialized $x_1$ randomly, and iterated inference until reaching energies within the lowest 10% of those observed on test images. For the combined discPC+genPC model, we required the generated samples to satisfy this low-energy criterion for both components. This set-up serves as a sampling process, allowing us to inspect images considered as highly likely by the models. As shown in Figure 6C, bPC produced realistic digits, while the combined model yielded poorly formed shapes. Quantitatively, bPC achieved higher Inception Scores ($6.05 \pm 0.17$ vs. $3.62 \pm 0.03$) and lower FID ($44.4 \pm 2.2$ vs. $140.5 \pm 2.1$). Similar trends held for BP-based baselines (see SM F).

These results clarify the observations in Section 4.1. discPC fails at generation because it produces broad, overconfident minima that admit many nonsensical and OOD inputs, leading to the noisy conditional generations in Figure 3 and in Sun & Orchard (2020). genPC, by contrast, learns narrow minima around class means, which rejects OOD inputs but identifies deviations from training-set mean as high-energy states, ultimately sacrificing discriminative precision. bPC integrates the two:

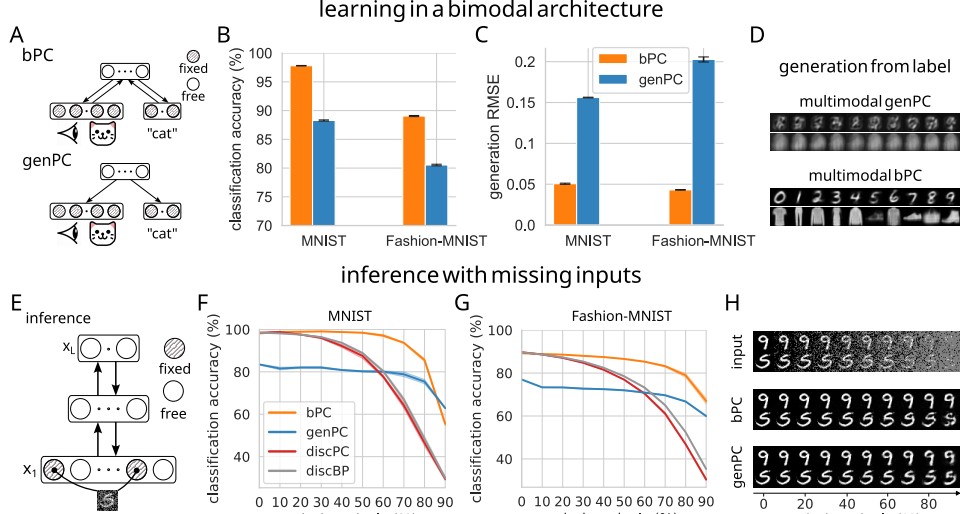

Figure 7: **bPC performance in a bimodal architecture and robustness against missing infor-mation.** A: Bimodal bPC and genPC. Note that bimodal bPC is fully equivalent to unimodal bPC in Section 4.1; it is bent here to serve visual purposes. B: Classification accuracy of bimodal PC models. C & D: Generation RMSE and examples. E: Model evaluation with partially occluded sensory inputs. F & G: Classification accuracy against percentage of missing pixels with MNIST and Fashion-MNIST respectively. H: Activity of $x_1$ after inference for genPC and bPC. Error bars and shaded region show s.e.m. across 5 seeds.

bidirectional predictions effectively regularize each other, sharpening discriminative minima while anchoring them to specific data points through the generative process. Shared latent layers thus prevent solutions overly biased to training mean or those that are overconfident, yielding superior performance in both classification and generation.

## 4.5 LEARNING IN BIOLOGICALLY RELEVANT TASKS

In this section, we demonstrate that bPC outperforms other predictive coding models in two biolog-ically relevant scenarios: (1) learning in a model architecture with two input streams analogous to two sensory modalities, and (2) classifying images from partially occluded inputs, similar to how human vision includes regions with missing sensory information, such as the retinal blind spot.

**Bimodal model architecture.** The brain often develops neural representations through associa-tions across modalities, such as linking spoken names to visual objects (Rosen et al., 2018). To test whether bPC can form such associations, we trained bPC and a bimodal variant of genPC on MNIST and Fashion-MNIST, with one latent layer connected to two inputs: an image and its one-hot label (Figure 7A; note that bPC is naturally bimodal and thus requires no restructuing). bPC incorporated both bottom-up and top-down connections for each modality, whereas genPC relied only on top-down pathways. After training, we evaluated cross-modal transfer by providing input to one modality and measuring inference quality in the other, classification accuracy for labels and RMSE for reconstructed images. As shown in Figure 7B-D, bPC significantly outperformed bimodal genPC on both tasks. This result is consistent with Sections 4.1: bPC's bidirectional pathways nat-urally support associative coding, while genPC must be restructured to handle multimodal inputs, where one pathway predicts the image (similar to genPC) and the other predicts the label (similar to discPC) and inherits the weaknesses of both genPC and discPC.

**Robustness to missing information.** The cortex can recognize objects even when sensory input is incomplete or occluded (Komatsu, 2006). We therefore tested the classification performance of the trained models from Section 4.1 (bPC, genPC, discPC, and discBP) under progressive occlu-sion of MNIST and Fashion-MNIST inputs, masking up to 90% of pixels (Figure 7E). Observed pixels were clamped in $x_1$, while missing ones were left free and initialized to zero. We then run inference on the missing pixels and in latent layers, with extended iterations to accommodate slower convergence. Results (Figure 7F-G) show that bPC maintained high classification accuracy even

with 80% missing pixels. genPC also retained some robustness but at consistently lower accuracy. In contrast, discPC and discBP collapsed beyond 50% occlusion. Visualizations in Figure 7H reveal that bPC actively reconstructed missing inputs via top-down priors, integrating them with observed evidence. genPC shared this generative capacity but lacked strong discriminative accuracy, while purely discriminative models could not compensate for missing information at all.

### 4.6 GENERATIVE AND DISCRIMINATIVE PROCESSING IN DEEP MODELS

To investigate whether bPC scales effectively to deeper models and more complex datasets, we trained bPC, discPC and discBP using the following architecture-dataset combinations: VGG-5 on CIFAR-10, VGG-9 on CIFAR-100, and VGG-16 on Tiny-ImageNet. After training, we evaluated discriminative performance by measuring the model's classification accuracy. Additionally, we assessed generative capabilities by evaluating classification accuracy when 30% and 50% of the input pixels were missing, following the methodology outlined in Section 4.5. To efficiently simulate the predictive coding models in this experiment, we employed error optimisation (Goemaere et al., 2025). This approach prevents energy decay in predictive coding models and enables the training of larger architectures. Refer to SM I for more implementation details.

Figure 8 confirms that bPC successfully combines discriminative and generative processing in deep networks. bPC achieves a classification accuracy comparable to discPC and discBP when full images are presented and significantly outperforms discPC and discBP on images with missing input information. For example, when 50% of pixels are missing, bPC classifies more than 60% more accurately than discPC and discBP on CIFAR-10 (Figure 8A). This improvement stems from bPC's generative processing, which fills in missing information. Figure 8D illustrates this effect: after inference, the activity of the input layer $x_1$ reveals that neurons lacking inputs have been updated to predict the missing values. Once the missing information is inferred, bPC classifies images accurately. Overall, even in larger models, bPC effectively balances discriminative and generative performance by jointly minimising bottom-up and top-down prediction errors.

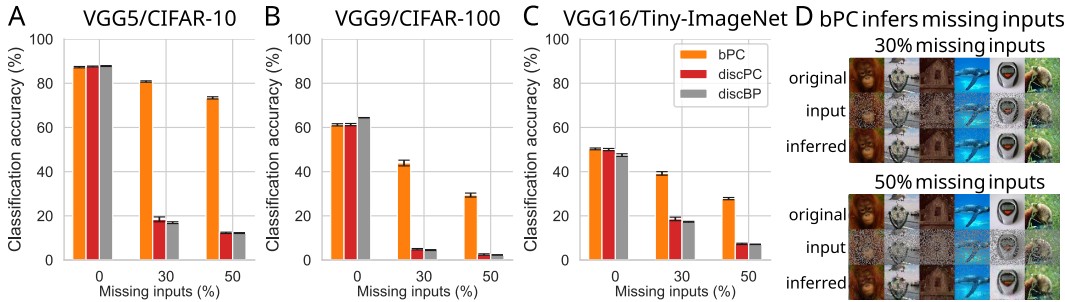

Figure 8: **bPC's discriminative and generative properties scale to deep networks.** A & B & C: Classification accuracy against the percentage of missing pixels for bPC, discPC and discBP on the architecture-dataset combinations: VGG-5 on CIFAR-10, VGG-9 on CIFAR-100, and VGG-16 on Tiny-ImageNet. D: Activity of input neurons $x_1$ after inference for bPC trained on Tiny-ImageNet. Error bars show s.e.m. across 5 seeds.

## 5 CONCLUSION

Inspired by empirical and theoretical insights into visual processing, we propose bidirectional predictive coding, a biologically plausible computational model of visual inference that integrates generative and discriminative processing. We demonstrate that bPC performs effectively across both supervised classification and unsupervised representation learning tasks, consistently outperforming or matching traditional predictive coding models. Our experiments reveal that the performance of bPC emerges from its ability to develop an energy landscape optimized simultaneously for both discriminative and generative tasks, thereby improving its robustness to out-of-distribution data. Furthermore, we show bPC's effectiveness in biologically relevant scenarios such as multimodal integration and inference with partially missing inputs. Overall, bPC offers a hypothesis for how flexible inference could emerge in the brain, while also providing a method to enhance the robustness of discriminative models in machine learning applications.

## REPRODUCIBILITY STATEMENT

A detailed description of all experiments is provided in the first section of the supplementary materials. The complete anonymised codebase is also attached to the supplementary materials to facilitate the reproducibility of our results.

## ACKNOWLEDGEMENTS

This work has been supported by the Medical Research Council UK grants MC_UU_00003/1, UKRI/MR/B000936/1 and by the Wellcome Trust grant 313955/Z/24/Z.

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

SUPPLEMENTARY MATERIAL

The supplementary material is organised as follows:

We begin with a detailed explanation of the methods underlying the experiments presented in the main paper. Next, we provide summary tables containing the exact values plotted in the figures. We then explore the learning performance of three additional predictive coding models. Following that, we examine the effect of scaling the discriminative and generative energies in bPC on both supervised and unsupervised learning performance. We also analyse the impact of removing max-pooling on generative artefacts. We present samples generated by a combined discBP and genBP model. After, we perform parameter count matched experiments to validate our results. We then discuss the relationship between hybridPC and bPC. Finally, we evaluate whether bPC learns effectively in deep models.

## A  DETAILS OF EXPERIMENTS

Here, we detail the training setup and the evaluation procedure of the experiments in the paper. The code with all models and experiments will be available upon publication . Our implementation of the predictive coding models was adapted from Pinchetti et al. (2025a).

### A.1  DATASETS

We evaluate the models on four standard image classification benchmarks: MNIST(LeCun et al., 2010), Fashion-MNIST(Xiao et al., 2017), CIFAR-10, and CIFAR-100(Krizhevsky, 2009a). Below, we summarise their properties and how they were subdivided in our experiments for training, validation and testing. The images of all datasets were normalised so that pixel values varied between $-1$ and 1.

- **MNIST**: A dataset of grayscale handwritten digits (0–9).

- **Fashion-MNIST**: A dataset of grayscale images of clothing items.

- **CIFAR-10**: A dataset of colour images categorised into 10 classes of everyday objects.

- **CIFAR-100**: Similar to CIFAR-10, but with 100 fine-grained object categories.

Table 1: Summary of dataset characteristics. The evaluation set of each dataset is split 50/50 into validation and test subsets.

| Dataset | Image Size | Channels | Classes | Train | Validation | Test |
|---|---|---|---|---|---|---|
| MNIST | $28 \times 28$ | 1 | 10 | 60,000 | 5,000 | 5,000 |
| Fashion-MNIST | $28 \times 28$ | 1 | 10 | 60,000 | 5,000 | 5,000 |
| CIFAR-10 | $32 \times 32$ | 3 | 10 | 50,000 | 5,000 | 5,000 |
| CIFAR-100 | $32 \times 32$ | 3 | 100 | 50,000 | 5,000 | 5,000 |

In addition to these benchmarks, we also consider the XOR learning task. XOR is a simple, non-linearly separable binary classification problem, often used to test the capacity of neural networks to learn non-linear functions. The input-output relationship of XOR is given in Table 2.

Table 2: XOR truth table with scaled inputs.

| Input 1 | Input 2 | Output (XOR) |
|---|---|---|
| -1 | -1 | -1 |
| -1 | 1 | 1 |
| 1 | -1 | 1 |
| 1 | 1 | -1 |

## A.2 MODEL ARCHITECTURES

We consider five different neural architectures in our experiments, selected based on the dataset and the learning setting.

**MLP for MNIST and Fashion-MNIST** For MNIST and Fashion-MNIST in Sections 4.1, 4.2 and 4.3, we use a network with neural layers of dimension [784, 256, 256, latent_size]. The latent layer size is set to 10 for supervised experiments, 30 for unsupervised experiments, and 40 for experiments with partial clamping of the top-layer activity ($x_4$). Both generative and discriminative predictions between layers follow the form $f(Wx + b)$, where $f$ is a nonlinearity, $W$ is a weight matrix, $b$ is a bias vector, and $x$ is the activation of the previous layer. This is the transformation of a layer in a multilayered perceptron (MLP). Consequently, unidirectional models (discPC, genPC, discBP, genBP) have the architecture of an MLP. hybridPC and bPC have the same number of parameters as two MLPs (one generative and one discriminative).

**CNN for unsupervised learning with CIFAR-10 and CIFAR-100** In the unsupervised representational learning experiments (Section 4.2) on CIFAR-10 and CIFAR-100, we use a model architecture with convolutions and a representational layer $x_L = x_6$ of 256 neurons. In this model, discriminative predictions consist of 5 strided convolution layers, with a stride of two, and one MLP layer. Generative predictions mirror the discriminative layers with 5 transposed convolution layers, with a stride of two, and one MLP layer. The convolutional-based layers perform the transformation $f(c(x))$, where $c$ is a (transposed) convolution layer and $f$ is a nonlinearity. A full description of the convolution-based layers can be found in Table 3.

**VGG-5 for combined supervised and unsupervised learning with CIFAR-10** For the CIFAR-10 experiments that combine supervised and unsupervised representational learning in Section 4.3, we use a VGG-5-style network with a 266-dimensional latent layer $x_L = x_6$ (10 neurons for class labels and 256 for representations). In this model, discriminative predictions consist of 4 convolutional layers with max pooling and one MLP layer. The generative predictions consist of 4 transposed convolution layers and one MLP layer. The discriminative convolution layers have a stride of one, and the transpose convolutions have a stride of two to compensate for the dimensionality reduction of the max-pooling. A full description of the convolution-based layers can be found in Table 3.

For the above three models, the activation function used in the discriminative prediction from $x_{L-1}$ to $x_L$ is an identity layer (no activation). The generative prediction from layer $x_2$ to $x_1$ uses a $tanh(\cdot)$ activation to constrain outputs to the range [-1, 1], matching normalised pixel values. The other activation functions are the same for the whole network and are determined using a hyperparameter search.

**MLP for XOR Task** For the XOR task in Section 4.4, we use a network with neural layers of dimension $[2, 16, 16, 1]$. The generative and discriminative predictions are equivalent to the ones used for MNIST and Fashion-MNIST, with the exception that: a sigmoid activation is used for the prediction from $x_3$ to $x_4$ and an identity activation is used for the prediction from $x_2$ to $x_1$. These changes stabilise training and help avoid bias because the tanh activation used for (Fashion-)MNIST saturates near ±1, but never reaches these values, causing the model to systematically under-predict XOR inputs.

**MLP for Bimodal Learning Task** In the bimodal generation task of Section 4.5, the network consists of two input layers, with 784 (image) and 10 (label) neurons, and one latent layer with 256 neurons. The predictions between the latent layer and the input layers are MLP layers. The bimodal bPC model is equivalent to the bPC model trained on (Fashion-)MNIST in Section 4.1, except it only has one hidden layer. In contrast, the bimodal genPC is different from the models considered in Section 4.1, and it can be re-interpreted as a unidirectional model with one generative MLP layer and one discriminative MLP layer. For fair comparison, we include an additional nonlinearity in the prediction layers of the genPC variant. The MLP layers become $f_1(Wf_2(x) + b)$. For the image modality, $f_1 = \tanh(\cdot)$; for the label modality, $f_1$ is the identity. The additional nonlinearity, $f_2$, ensures parity with bPC models, which include nonlinear discriminative projections.

Table 3: Detailed architectures of convolution-based models. Convolution and transposed convolution have the same kernel sizes and paddings. The values in the brackets give the parameter value for each convolution-based layer in the model, starting from the layer closest to the input image.

|  | **CNN** | **VGG-5** |
| --- | --- | --- |
| Channel Sizes | [32, 64, 128, 256, 512] | [128, 256, 512, 512] |
| Kernel Sizes | [3, 3, 3, 3, 3] | [3, 3, 3, 3] |
| Paddings | [1, 1, 1, 1,1] | [1, 1, 1, 1] |
| Strides conv. | [2, 2, 2, 2,2] | [1, 1, 1, 1] |
| Strides transposed conv. | [2, 2, 2, 2,2] | [2, 2, 2, 2] |
| Output Paddings transposed conv. | [1, 1, 1, 1,1] | [0, 0, 0, 0] |
| Pooling Window | - | $2 \times 2$ |
| Pooling Stride | - | 2 |

---

**Algorithm 1:** Training procedure of PC models.

---

**Require:** Model with neural activities $x$, parameters $\theta$, energy $E$ and initialisation function $init(\cdot)$. Dataset $\{y_p\}_{p=1}^P$ with $P$ mini-batches of $B$ elements. Number of epochs $N$. Activity optimiser $optim_x(\cdot)$, number of activity updates $K$, and parameter optimiser $optim_\theta(\cdot)$

**for** $n = 1$ **to** $N$ **do**
  **for** $p = 1$ **to** $P$ **do**
    // Independent inference for each sample in batch
    $x_b \leftarrow init(y_{p,b}), \quad 1 \leq b \leq B$
    **for** $k = 1$ **to** $K$ **do**
      $x_b \leftarrow optim_x(\frac{\partial E_b}{\partial x_b}), \quad 1 \leq b \leq B$
    // Sum of parameter updates for batch
    $\theta \leftarrow optim_\theta(\frac{1}{B} \sum_b^B \frac{\partial E_b}{\partial \theta})$

---

## A.3 TRAINING PROCEDURES

We followed the procedures outlined in Algorithm 1 to train predictive coding models and hybridBP, and the procedure outlined in Algorithm 2 for the remaining backpropagation models. For a given task, both algorithms were trained on the same dataset for the same number of epochs.

**Optimisers**

The optimisation procedures differ between activity updates and parameter updates. In Algorithm 1, neural activities are updated using stochastic gradient descent with momentum. This optimiser outperforms Adam in PC models (Pinchetti et al., 2025a). In contrast, Algorithm 2 does not include activity updates and only optimises parameters. For parameter updates in both algorithms, we use the

---

**Algorithm 2:** Training procedure of BP models.

---

**Require:** Model with parameters $\theta$ and forward pass $fp(\cdot, \theta)$. Energy function $E$. Dataset $\{y_p\}_{p=1}^P$ with $P$ mini-batches of $B$ elements. Number of epochs $N$. Parameter optimiser $optim_\theta(\cdot)$

**for** $n = 1$ **to** $N$ **do**
  **for** $p = 1$ **to** $P$ **do**
    // Independent forward pass for each sample in batch
    $\hat{y}_b \leftarrow fp(y_{p,b}, \theta), \quad 1 \leq b \leq B$
    // Sum of parameter updates for batch
    $\theta \leftarrow optim_\theta(\frac{1}{B} \sum_b^B \frac{\partial E(\hat{y}_b, y_{p,b})}{\partial \theta})$

---

AdamW optimiser (Loshchilov and Hutter, 2017). The optimiser hyperparameters were determined via a hyperparameter search.

**Initialisation functions**

The initialisation procedure in Algorithm 1 generally consists of (1) clamping the activity $x_1$ and $x_L$ to images and labels depending on the learning task and (2) performing a feedforward sweep. This sweep initialises the activity of each layer by propagating inputs through the model in the direction of prediction. For example, a bottom-up feedforward sweep for a three-layer model initialises the second and third layers as: $x_2 = f(x_1)$ and $x_3 = V_2 f(V_1 f(x_1))$. This approach, originally proposed by Whittington and Bogacz (2017), significantly reduces the time required for inference and improves learning. The following models use bottom-up feedforward initialisation: discPC, hybridPC, bPC, and hybridBP. The genPC model, by contrast, uses a top-down feedforward initialisation. The difference between hybridBP and the other models initialised using a bottom-up sweep is that hybridBP contains a single layer of neural activities, $x_L$, that can be initialised using a full bottom-up forward pass.

The bimodal genPC model from Section 4.5 differs from the other PC models in its initialisation procedure. Because there are no direct prediction paths from the two modality-specific input layers to the shared latent layer, a standard feedforward sweep cannot be used to initialise the latent layer. Instead, we tested two alternatives for initialising the latent layer: zero-initialisation and Xavier uniform initialisation. The best initialisation depended on the task and was determined with a hyperparameter search. For the bimodal bPC model, we retained the feedforward sweep used for other bPC models, using a forward sweep from the image modality to the latent layer.

**Energy functions**

The energy functions for all PC and BP models are listed in Table 4. For PC models, the energy functions follow the formulations described in the main text, except that we use a generalised notation: $f_l$ denotes the transformation applied at layer $l$, allowing for models beyond standard MLPs. BP model energy functions use squared errors to remain consistent with PC formulations.

The hybridBP model includes a stop-gradient operation in its energy function. This ensures that its second loss term is only applied during parameter updates, not iterative inference. This loss term objective is to improve the discriminative initialisation of the model's latent layer. In the broader machine learning literature, hybridBP can be viewed as a generative model that performs Bayesian inference by combining amortised inference with iterative refinement of the latent variables (Marino et al., 2018).

In the case of the autoencoder (AE), the energy function departs from standard autoencoders to accommodate experiments described in Section 4.3, where part of the latent state $x_L$ is fixed to class labels. The AE energy includes an additional term: $\|x_{L,ae} - f_{p,disc}(x_1, V)\|_2^2$ where $x_{L,ae} := [x_{L,0:k-1}, f_{p,disc}(x_1, V)_{k:D}]$ that combines $k$ fixed neurons and discriminative predictions. This term penalises the error between the fixed components of $x_{L,ae}$ and their prediction from the discriminative part of the model. Notably, this term is only non-zero for the fixed neurons. The additional energy term is also equal to zero in the representational learning experiments of Section 4.2 because none of the neurons of $x_{L,ae}$ are fixed. For reference, the discriminative part of our AE is usually referred to as the encoder and the generative part as the decoder.

In the combined supervised and unsupervised learning task (Section 4.3), we apply separate energy scaling to different parts of the models. Unsupervised learning benefits from equal weighting of discriminative and generative energies, while supervised tasks perform better when the generative energy is down-scaled. To balance these, we set $\alpha_{disc}$ of the free (unclamped) neurons in $x_L$ to $\alpha_{gen}$. This allows for the generative energy to be down-scaled in the whole network except for the free neurons of $x_L$ that specialise in unsupervised representational learning. This adjustment was used for bPC, hybridBP, and AE, but not for hybridPC. In hybridPC, the discriminative weights predicting free and fixed neurons are independent due to the use of local learning rules. Moreover, these discriminative weights have minimal influence on the generative weights, as inference is fully driven by the generative energy. As a result, scaling the discriminative and generative energies in different parts of the model merely scales the corresponding gradients. This effect has no impact on learning because the AdamW optimiser normalises gradients. In contrast, in hybridBP, some parameters are shared in predicting both free and fixed neurons. This introduces a dependency between the scaling

of energies associated with free and fixed neurons in $x_L$. Therefore, we introduced explicit scaling factors $\alpha_{disc}$ and $\alpha_{gen}$ in hybridBP's energy function for this task.

| Model | Loss Function |
|---|---|
| discPC | $E_{\mathrm{disc}}(x, V) = \sum_{l=2}^{L} \frac{1}{2}\|x_l - f_l(x_{l-1}, V_{l-1})\|_2^2$ |
| discBP | $\mathcal{L}_{disc}(x_1, x_L, V) = \frac{1}{2}\|x_L - f_{p,disc}(x_1, V)\|_2^2$ |
| genPC | $E_{\mathrm{gen}}(x, W) = \sum_{l=1}^{L-1} \frac{1}{2}\|x_l - f_l(x_{l+1}, W_{l+1})\|_2^2$ |
| genBP | $\mathcal{L}_{gen}(x_1, x_L, W) = \frac{1}{2}\|x_1 - f_{p,gen}(x_L, W)\|_2^2$ |
| hybridPC | $E_{\mathrm{hybrid}}(x, W, V) = \sum_{l=1}^{L-1} \frac{1}{2}\|x_l - f_l(x_{l+1}, W_{l+1})\|_2^2 + \sum_{l=2}^{L} \frac{1}{2}\|\mathrm{sg}(x_l) - f_l(\mathrm{sg}(x_{l-1}), V_{l-1})\|_2^2$ |
| hybridBP | $\mathcal{L}_{\mathrm{hybrid}}(x_1, x_L, W, V) = \frac{1}{2}\|x_1 - f_{p,gen}(x_L, W)\|_2^2 + \frac{1}{2}\|\mathrm{sg}(x_L) - f_{p,disc}(x_1, V)\|_2^2$ |
| bPC | $E(x, W, V) = \sum_{l=1}^{L-1} \frac{\alpha_{\mathrm{gen}}}{2}\|x_l - f(x_{l+1}, W_{l+1})\|_2^2 + \sum_{l=2}^{L} \frac{\alpha_{\mathrm{disc}}}{2}\|x_l - f(x_{l-1}, V_{l-1})\|_2^2$ |
| AE | $\mathcal{L}(x_1, x_L, W, V) = \frac{\alpha_{\mathrm{gen}}}{2}\|x_1 - f_{p,gen}(x_{L,ae}, W)\|_2^2 + \frac{\alpha_{\mathrm{disc}}}{2}\|x_{L,ae} - f_{p,disc}(x_1, V)\|_2^2,$ $x_{L,ae} := [x_{L,0:k-1}, f_{p,disc}(x_1, V)_{k:D}]$ |

Table 4: Energy functions for the predictive coding and backpropagation models considered in the paper. In the PC models, $f_l(x, \theta)$ is the transformation between layers with input $x$ and parameters $\theta$. In the BP models, $f_p$ denotes a forward pass and $x_{L,ae}$ is an activity vector composed of $k$ fixed neurons and fills the remaining outputs of the discriminative forward pass of the AE.

## A.4 TRAINING HYPERPARAMETERS

In this Section, we report the hyperparameter search space for the experiments in Sections 4.1, 4.2, 4.3, and 4.5, as well as the training parameters for the models in Section 4.4. All hyperparameter tuning was performed using Bayesian optimisation from Weights and Biases (Biewald, 2020).

Tables 5, 6, and 7 list the hyperparameter search spaces for the models trained in Sections 4.1, 4.2, and 4.3, respectively. The hyperparameter search was conducted using the validation sets of each dataset. In Section 4.1, discPC and genPC were tuned for classification accuracy and generation RMSE separately, while bPC and hybridPC were jointly tuned on both metrics. This ensures a fair comparison, as bPC and hybridPC have twice as many parameters as discPC and genPC. For joint tuning, we combined the two metrics using the objective $2 \cdot (1 - \mathrm{accuracy}/100) + \mathrm{RMSE}$. In Section 4.2, the models were tuned for the image reconstruction MSE, linear decoding accuracy and generation FID separately. For the reconstruction, the RMSE was reported instead of the MSE for consistency with the generation RMSE. In Section 4.3, all models were jointly optimised for classification accuracy and reconstruction MSE. We combined the two metrics using $2 \cdot (1 - \mathrm{accuracy}/100) + \mathrm{MSE}$ for MNIST and Fashion-MNIST, and $(1 - \mathrm{accuracy}/100) + 4 \cdot \mathrm{MSE}$ for CIFAR-10. The RMSE was also reported after tuning instead of the MSE. The weightings in the combined objectives compensate for the scale difference between the metrics.

Table 8 reports the training parameters for the models used to generate MNIST samples in Section 4.4. The leaky ReLU activation function was used because it generated the best samples across model types, and the other parameters were set to default values that ensured stable learning across models.

Table 9 lists the hyperparameter search space for the bimodal bPC and genPC models trained in Section 4.5. The genPC model was tuned separately for classification accuracy and reconstruction RMSE, while the bPC model was tuned jointly on both metrics. The joint tuning followed the same combined metric as described above for MNIST. During training and evaluation, more inference steps were used for genPC than bPC, as genPC lacks an effective feedforward initialisation scheme, which slows inference convergence.

Table 5: Hyperparameter search configuration for experiments in Section 4.1 for both MNIST and Fashion-MNIST datasets.

| Parameter | bPC | hybridPC | discPC | genPC | genBP | discBP |
|---|---|---|---|---|---|---|
| Epoch | | | 25 | | | |
| Batch Size | | | 256 | | | |
| Activation | | | [leaky relu, tanh, gelu] | | | |
| $lr_x$ | | $(1e\text{-}3, 5e\text{-}1)^2$ | | $(1e\text{-}4, 5e\text{-}2)^2$ | - | |
| momentum$_x$ | | [0.0, 0.5, 0.9] | | | - | |
| $lr_\theta$ | | $(1e\text{-}5, 1e\text{-}3)^2$ | | $(1e\text{-}6, 1e\text{-}4)^2$ | $(1e\text{-}5, 1e\text{-}3)^2$ | |
| weight_decay$_\theta$ | | | $(1e\text{-}5, 1e\text{-}2)^2$ | | | |
| T | | 8 | | | - | |
| T eval | | 100 | | | - | |
| $\alpha_{gen}$ | $1e[0,\text{-}4;\text{-}1]^1$ | | - | | - | |
| $\alpha_{disc}$ | 1 | | - | | - | |

[1]: "[a, b; c]" denotes a sequence of values from a to b with a step size of c.
[2]: "(a, b)" represents a log-uniform distribution between a and b.

Table 6: Hyperparameter search configuration for experiments in Section 4.2 for MNIST , Fashion-MNIST, CIFAR-10, and CIFAR-100 datasets. Parameters unique to the models trained on MNIST and Fashion-MNIST are indicated with MLP. Parameters unique to the CIFAR datasets are indicated with CNN.

| Parameter | bPC | genPC | hybridPC | hybridBP | AE |
|---|---|---|---|---|---|
| Epoch (MLP) | | | 25 | | |
| Epoch (CNN) | | | 50 | | |
| Batch Size | | | 256 | | |
| Activation | | | [leaky relu, tanh, gelu] | | |
| $lr_x$ | | | $(1e\text{-}3, 5e\text{-}1)^1$ | | - |
| momentum$_x$ | | | [0.0, 0.5, 0.9] | | - |
| $lr_\theta$ (MLP) | | | $(1e\text{-}5, 1e\text{-}3)^1$ | | |
| $lr_\theta$ (CNN) | | | $(1e\text{-}5, 1e\text{-}3)^{1,2}$ | | |
| weight_decay$_\theta$ | | | $(1e\text{-}5, 1e\text{-}2)^1$ | | |
| T (MLP) | | | 8 | | - |
| T (CNN) | | | 32 | | - |
| T eval | | | 100 | | - |
| $\alpha_{gen}$ | 1 | | - | - | 1 |
| $\alpha_{disc}$ | 1 | | - | - | 1 |

[1]: "(a, b)" represents a log-uniform distribution between a and b.
[2] Learning rates of $\theta$ were scaled with warmup-cosine-annealing scheduler without restart.

## A.5  EVALUATION

In our experiments, we consider six different evaluation procedures discussed below.

**Classification accuracy** We evaluate classification performance using accuracy, defined as the percentage of input images correctly classified by each model.

For the PC models, classification is performed by first setting $x_1$ to the input image. We initialise the remaining layers using a bottom-up feedforward sweep, followed by 100 steps of iterative inference. The predicted class is determined by identifying the neuron in the output layer $x_L$ (corresponding to class encoding units) with the highest activity. The bottom-up initialisation significantly reduces the time required to reach steady state during inference. However, bottom-up initialisation does not apply to genPC and bimodal genPC due to the absence of bottom-up predictive pathways. In genPC, we initialise $x_L$ to zero and perform a top-down feedforward sweep instead. For bimodal genPC, the latent layer is initialised identically to the procedure used during training.

For the BP models, classification is performed via a standard discriminative forward pass from the input image, with the predicted class taken as the one corresponding to the highest output activation.

Table 7: Hyperparameter search configuration for experiments in Section 4.3. Parameters unique to the models trained on MNIST and Fashion-MNIST are indicated with MLP. Parameters unique to the CIFAR-10 dataset are indicated with VGG. Two activity optimisers and two parameter optimisers are used for the VGG models. This allows the models to have different learning rates for different task-specific parts of the networks. One activity optimiser was used for the free neurons of $x_L$ with learning rate $lr_{x,\text{free}}$, and another for all other neurons with learning rate $lr_x$. Both activity optimisers use the same momentum parameter. One parameter optimiser with learning rate $lr_{\theta,gen}$ was used for the generative parameters and the discriminative parameters predicting the free neurons of $x_L$. The other parameter optimiser was used for the remaining discriminative parameters. Both parameter optimisers use the same weight decay.

| Parameter | bPC | hybridPC | hybridBP | AE |
|---|---|---|---|---|
| Epoch (MLP) | | 25 | | |
| Epoch (VGG) | | 50 | | |
| Batch Size | | 256 | | |
| Activation | | [leaky relu, tanh, gelu] | | |
| $lr_x$ | | (1e-3, 5e-1)[1] | | - |
| $lr_{x,free}$ (VGG) | | (1e-3, 5e-1)[1] | | - |
| $momentum_x$ | | [0.0, 0.5, 0.9] | | - |
| $lr_\theta$ (MLP) | | (1e-5, 1e-3)[1] | | |
| $lr_{\theta,disc}$ (VGG) | | (1e-5, 1e-3)[1,3] | | |
| $lr_{\theta,gen}$ (VGG) | | (1e-5, 1e-2)[1,3] | | |
| $weight\_decay_\theta$ | | (1e-5, 1e-2)[1] | | |
| T (MLP) | | 8 | | - |
| T (VGG) | | 32 | | - |
| T eval | | 100 | | - |
| $\alpha_{gen}$ (MLP) | 1e[0,-4;-1][1] | - | 1e[0,-4;-1][1] | 1e[0,-4;-1][1] |
| $\alpha_{gen}$ (VGG) | 1e[-4,-7;-1][1] | - | 1e[0,-8;-1][1] | 1e[0,-8;-1][1] |
| $\alpha_{disc}$ | 1 | - | 1 | 1 |

[1]: "(a, b)" represents a log-uniform distribution between a and b.
[2]: "[a, b; c]" denotes a sequence of values from a to b with a step size of c.
[3] Learning rates of $\theta$ were scaled with warmup-cosine-annealing scheduler without restart.

Table 8: Hyperparameters for models used to sample MNIST images in Section 4.1.

| Parameter | bPC | genPC | discPC |
|---|---|---|---|
| Epoch | | 25 | |
| Batch Size | | 256 | |
| Activation | | leaky relu | |
| $lr_x$ | | 0.01 | |
| $momentum_x$ | | 0.0 | |
| $lr_\theta$ | | 1e-4 | |
| $weight\_decay_\theta$ | | 5e-3 | |
| T | | 8 | |
| T eval | | 100 | |
| $\alpha_{gen}$ | 1e-4 | | - |
| $\alpha_{disc}$ | 1 | | - |

**Conditional generation of mean class images** We evaluate the generative performance of the models by computing the root mean squared error (RMSE) between images generated for each class and the corresponding class-average image. We obtain the class-average image by computing the average image across all images belonging to that class in the evaluation set. To compute the RMSE, we calculate the squared error for each pixel between the generated image and the average image for the corresponding class. These errors are averaged across all pixels and classes, and the square root of this mean is reported as the final RMSE.

Table 9: Hyperparameter search configuration for bimodal models in Section 4.5 for both MNIST and Fashion-MNIST datasets.

| Parameter | bPC | genPC |
|---|---|---|
| Epoch | 25 | |
| Batch Size | 256 | |
| Activation | [leaky relu, tanh, gelu] | |
| $lr_x$ | (1e-3, 5e-1)$^2$ | |
| $momentum_x$ | [0.0, 0.5, 0.9] | |
| $lr_\theta$ | (1e-5, 1e-3)$^2$ | |
| $weight\_decay_\theta$ | (1e-5, 1e-2)$^2$ | |
| T | 8 | 20 |
| T eval | 100 | 1000 |
| $\alpha_{gen}$ | 1e[0,-4;-1][1] | - |
| $\alpha_{disc}$ | 1 | - |

[1]: "[a, b; c]" denotes a sequence of values from a to b with a step size of c.
[2]: "(a, b)" represents a log-uniform distribution between a and b.

To generate an image from a PC model conditioned on a class label, we set the top-layer activity $x_L$ to the one-hot encoding of the target label. The remaining layers are then initialised using a top-down feedforward sweep. One exception is the discPC model, which does not support top-down initialisation. For discPC, we initialise the input layer $x_1$ to zero activity and then perform a bottom-up feedforward sweep. Following initialisation, we run 100 steps of iterative inference and extract the final activity of the input layer $x_1$ as the generated image.

For the BP models, we generate an image by performing a generative forward pass starting from a one-hot label vector.

**Image reconstruction from representations** We assess the quality of learned representations by measuring the RMSE between reconstructed images and their original inputs. RMSE is computed in the same manner as for conditional image generation.

For PC models and hybridBP, the reconstruction procedure is as follows: (1) clamp $x_1$ to the input image, (2) initialise the remaining layers via a bottom-up feedforward sweep, (3) perform 100 steps of iterative inference, (4) clamp $x_L$ to its activity after the inference, (5) re-initialise the other layers using a top-down feedforward sweep, (6) run 100 additional inference steps, and (8) record the final activity in $x_1$ as the reconstructed image.

For the autoencoder, an image is reconstructed by performing a discriminative forward pass using the input image to obtain a representation, followed by a generative forward pass from the representation to get the reconstructed image.

In Section 4.3, where labels are also provided during reconstruction, the above procedure is slightly modified. For PC models and hybridBP, certain neurons of $x_L$ are additionally clamped to the ground truth label during the first step. During the fourth step, the label components and the remaining latent representation (recorded after inference) are clamped in $x_L$. For the AE, we retain the standard discriminative forward pass, but replace the predicted label portion of its representation with the true label before passing it through the generative model to produce the reconstructed image.

**Linear readout/decoding accuracy** We further evaluate the quality of the learned representations by measuring their linear readout accuracy. The representations are obtained in the same way as for image reconstruction. A linear classifier head is then trained with backpropagation for each model to classify images based on their representations. This evaluation tests whether representations of different classes are linearly separable. Higher decoding accuracy reflects better representations.

**FID of generated image samples.** We compare the ability of bPC, genPC, and hybridPC models to learn probability distributions using stochastic extensions of predictive coding. Predictive coding can be equipped with stochastic neural dynamics to model distributions of sensory inputs in an unsupervised manner by injecting noise into the inference process. This idea was originally proposed in (Oliviers et al., 2024) for the genPC model, and here we apply it to genPC, hybridPC, and bPC.

When stochastic dynamics are applied, the noisy inference process can be shown to generate samples from the posterior distribution of the probabilistic model defined by predictive coding, conditioned on the sensory input. These posterior samples can then be used for parameter learning through Monte Carlo Expectation Maximisation.

The training procedure mirrors that of the deterministic experiments: each iteration consists of several (now noisy) activity updates followed by a single parameter update. This is equivalent to using a single posterior sample per parameter update, a common practice in related generative models such as variational autoencoders.

To accelerate inference, we further incorporate momentum into the neural dynamics, as introduced for predictive coding in (Pinchetti et al., 2025b). The resulting dynamics correspond to a discrete-time second-order Langevin process. In practice, this is implemented by adding Gaussian noise to the gradients of the energy function and updating activities with stochastic gradient descent with momentum:

$$\Delta x_l = lr_x r_l, \tag{7}$$

$$\Delta r_l = -lr_x \nabla_{x_l} E; -; lr_x(1-m)r_l; +; \sqrt{2(1-m)lr_x}, N. \tag{8}$$

After training, the models can be used to generate input samples. For a top-down predictive coding model, this is done by ancestral sampling: first sampling the top latent layer $x_L \sim N(0, I)$, and then recursively sampling each lower layer from its conditional Gaussian distribution $N(x_l; f_l(x_{l+1}, W_{l+1}), I)$. The prior distribution on $x_L$ emerges from the additional activity decay applied to $x_L$ for the unsupervised learning experiments.

To evaluate generative performance, we compute the Fréchet Inception Distance between generated and real samples. We use the open-source library pytorch-fid (Seitzer, 2020a), adapted to MNIST by replacing the standard Inception network with a ResNet-18 trained on MNIST as the feature extractor.

As a baseline, we include a variational autoencoder (VAE) trained on the same learning task. The VAE architecture and parameter count are matched to those of hybridPC and bPC to enable a fair comparison. Training follows the standard VAE procedure: the encoder network approximates the posterior distribution over latent variables, while the decoder parametrises the generative model. The model is optimised end-to-end via backpropagation using the standard VAE objective.

**Visualisation of Energy Landscapes on XOR** We visualise the energy landscapes of trained genPC, discPC, and bPC models on the XOR task. To generate the landscape, we clamp the input layer neurons to a 2D coordinate within the range [-3,3], sampled at intervals of 0.25 along both axes. Simultaneously, we clamp the output layer $x_L$ to the one-hot encoding of one of the two classes. For each coordinate–label combination, we run 10,000 steps of iterative inference to ensure convergence to equilibrium and record the final energy of the model. This procedure is repeated over the grid of 2D inputs and both class labels, allowing us to plot the full energy landscape. For fair comparison, we also include the combined energy landscape of discPC and genPC. Together, the combined parameter count of discPC and genPC matches the total parameter count of bPC. Equal weighting is applied to the energy values of the discPC and genPC models when adding their energy, as we use equal weighting for the discriminative and generative components in the bPC model ($\alpha_{disc} = \alpha_{gen} = 1$).

**MNIST image generation with PC models** We evaluate the generative capabilities of the bPC model and a combined genPC and discPC model on the MNIST dataset. The goal is to assess how likely these models are to assign low energy to implausible, label-inconsistent samples.

As a baseline, we first estimate the energy distribution over the test set for each model. For each test image, we clamp $x_1$ to the image and $x_L$ to its associated label. The model is then initialised as during training, and we run 50,000 steps of iterative inference to reach a steady state. We compute the 10th percentile from the resulting energy values as an energy threshold for high-quality, in-distribution samples. This process was repeated independently for bPC, genPC and discPC.

To generate images conditioned on a label, we use the following procedure: (1) randomly initialise the input layer $x_1$ by sampling each neuron's activity uniformly from the interval [-1,1], (2) clamp $x_L$ to the target label, (3) initialise all hidden layers to zero, (4) run iterative inference until the median energy of the batch falls below the previously determined 10th percentile threshold, (5)

retain the 50% of samples within the batch that fall below or equal to the energy threshold and discard the rest. Using this method, we generate 256 samples per model.

In the combined genPC and discPC model, the input layer $x_1$ is shared by both models, and its updates during inference are influenced by both energy functions. However, genPC typically exhibits energy magnitudes significantly larger than those of discPC. To prevent one component from dominating the updates to $x_1$, we scale the generative energy by a factor equal to the ratio of the discPC energy threshold to the genPC energy threshold. Additionally, inference is only terminated once the generative and the discriminative energies fall below their respective thresholds.

The models used for sample generation are trained separately from those used in the supervised experiments from Section 4.1. During preliminary testing, we found that generative quality strongly depends on the choice of activation function. In particular, leaky-ReLU yielded superior image samples. As a result, we train all models using leaky-ReLU activations and a shared set of training hyperparameters. These parameters are selected for their stability and effectiveness across all model types for classification and generation tasks.

We assess the quality and diversity of the generated samples using two standard metrics:

- Fréchet Inception Distance (FID): The FID measures the distance between the distributions of real and generated images (Heusel et al., 2017). We use the public implementation of FID from Seitzer (2020b), but modify it to use a ResNet classifier trained on MNIST, instead of the original Inception model trained on ImageNet. This ensures that the FID score better reflects visual quality and diversity on the MNIST domain.

- Inception Score (IS): The Inception Score evaluates how easily a classifier can identify the class of a generated image (Salimans et al., 2016). We used a publicly available implementation for the MNIST dataset (Chen, 2020).

**Classification accuracy with partially missing inputs**

To assess model robustness to missing input data, we evaluate classification accuracy under varying levels of input occlusion. This evaluation follows the same procedure as standard classification (described above), but for images missing a random subset of pixels. The proportion of missing pixels ranges from 10% to 90%, in increments of 10%. Missing pixels are selected uniformly at random, independently of their spatial location within the image. For all models, missing pixels in $x_1$ are initialised to zero activity and are left unclamped, allowing the model to update their values during inference. We repeat the same classification procedure as before, but we increased the number of inference steps to 600,000 to ensure convergence. This is necessary because predictive coding models converge more slowly when input information is incomplete (Tang et al., 2023). In addition to measuring classification accuracy, we also record the post-inference activity of neurons in $x_1$ to qualitatively assess the model's ability to fill in the missing input.

### A.6  COMPUTE RESOURCES

All experiments were conducted on NVIDIA RTX A6000 GPUs. Training an MLP model for MNIST and Fashion-MNIST experiments across tasks took less than one minute. Training unsupervised learning models on CIFAR-10 and CIFAR-100 of Section 4.2 took approximately 15 minutes. Training the combined supervised and unsupervised models of Section 4.3 took approximately one hour and 15 minutes. The majority of the compute was spent on hyperparameter tuning. The total training time for hyperparameter tuning of the models of Section 4.1 is $\pm 50$h. The total training time for hyperparameter tuning of the models of Section 4.2 is $\pm 30$h for MNIST and Fashion-MNIST and $\pm 400$h for CIFAR-10 and CIFAR-100. The total training time for hyperparameter tuning of the models of Section 4.3 is $\pm 25$h for MNIST and Fashion-MNIST and $\pm 750$h for CIFAR-10. The total training time for the tuning of the bimodal models of Section 4.5 is $\pm 20$h.

### A.7  LLM USE

ChatGPT Edu was used to polish writing.

# B    RESULTS

Tables 10 to 15 report the result illustrated in Figures 3, 4, 5, and 7. These results are obtained on the test set of the datasets for five different weight initialisations.

Table 10: Classification accuracy and class average image generation for models considered in Section 4.1. Higher accuracy and lower RMSE are better. We report the mean +/- sem over five seeds. Results indicated with $*$ are significantly worse in performance than bPC determined using an independent-samples t-test (n=5, $p < 0.05$).

| Model | Acc % | | RMSE | |
|---|---|---|---|---|
| | **MNIST** | **Fashion-MNIST** | **MNIST** | **Fashion-MNIST** |
| bPC | $98.10^{\pm 0.05}$ | $89.24^{\pm 0.12}$ | $0.0581^{\pm 0.0004}$ | $0.0415^{\pm 0.0005}$ |
| hybridPC | $86.22^{\pm 0.15*}$ | $80.34^{\pm 0.11*}$ | $0.0612^{\pm 0.0003}$ | $0.0480^{\pm 0.0016}$ |
| genPC | $83.48^{\pm 0.21*}$ | $77.00^{\pm 0.16*}$ | $0.0198^{\pm 0.0001}$ | $0.0140^{\pm 0.0001}$ |
| discPC | $98.43^{\pm 0.01}$ | $89.74^{\pm 0.14}$ | $0.3133^{\pm 0.0224*}$ | $0.3326^{\pm 0.0024*}$ |
| BP | $98.48^{\pm 0.10}$ | $89.66^{\pm 0.11}$ | $0.0198^{\pm 0.0001}$ | $0.0128^{\pm 0.0001}$ |

Table 11: Image reconstruction RMSE from latent representations for models considered in Section 4.2. Lower RMSE is better. We report the mean +/- sem over five seeds. Results indicated with $*$ are significantly worse in performance than bPC determined using an independent-samples t-test (n=5, $p < 0.05$).

| Model | MNIST | Fashion-MNIST | CIFAR-10 | CIFAR-100 |
|---|---|---|---|---|
| bPC | $0.2320^{\pm 0.0010}$ | $0.2497^{\pm 0.0004}$ | $0.1311^{\pm 0.0005}$ | $0.1366^{\pm 0.0007}$ |
| genPC | $0.2473^{\pm 0.0020*}$ | $0.2868^{\pm 0.0013*}$ | $0.1837^{\pm 0.0009*}$ | $0.2077^{\pm 0.0003*}$ |
| hybridPC | $0.2401^{\pm 0.0012*}$ | $0.2508^{\pm 0.0007}$ | $0.1664^{\pm 0.0015*}$ | $0.2089^{\pm 0.0071*}$ |
| AE | $0.1565^{\pm 0.0006}$ | $0.1868^{\pm 0.0001}$ | $0.1135^{\pm 0.0050}$ | $0.1171^{\pm 0.0042}$ |
| BP | $0.1969^{\pm 0.0004}$ | $0.2084^{\pm 0.0002}$ | $0.0964^{\pm 0.0004}$ | $0.0983^{\pm 0.0002}$ |

Table 12: Linear decoding accuracy (%) across datasets for different models considered in Section 4.2. Higher is better. We report the mean $\pm$ sem over five seeds. Results indicated with $*$ are significantly worse in performance than bPC determined using an independent-samples t-test (n=5, $p < 0.05$).

| Model | MNIST | Fashion-MNIST | CIFAR-10 | CIFAR-100 |
|---|---|---|---|---|
| bPC | $89.99^{\pm 0.01}$ | $81.82^{\pm 0.01}$ | $50.52^{\pm 0.01}$ | $60.68^{\pm 0.01}$ |
| genPC | $86.93^{\pm 0.01*}$ | $80.44^{\pm 0.01*}$ | $48.23^{\pm 0.02*}$ | $50.45^{\pm 0.02*}$ |
| hybridPC | $89.07^{\pm 0.01*}$ | $81.59^{\pm 0.01}$ | $49.61^{\pm 0.01*}$ | $59.31^{\pm 0.01*}$ |
| AE | $90.76^{\pm 0.01}$ | $83.31^{\pm 0.01}$ | $47.29^{\pm 0.02}$ | $63.78^{\pm 0.04}$ |
| hybridBP | $93.44^{\pm 0.01}$ | $78.69^{\pm 0.018*}$ | $50.63^{\pm 0.01}$ | $85.14^{\pm 0.01}$ |

Table 13: FID scores for different models trained with 50 and 250 activity updates before each weight update considered in Section 4.2. Lower is better. We report the mean $\pm$ sem over five seeds. The VAE only has one value because it does not have iterative inference.

| Model | FID @ 50 updates | FID @ 250 updates |
|---|---|---|
| bPC | $5.21^{\pm 0.26}$ | $3.34^{\pm 0.53}$ |
| genPC | $7.86^{\pm 1.27}$ | $4.56^{\pm 0.30}$ |
| hybridPC | $5.01^{\pm 0.30}$ | $4.28^{\pm 0.32}$ |
| VAE | $5.79^{\pm 0.21}$ | |

Table 14: Classification accuracy and image reconstruction RMSE from latent representations for models considered in Section 4.3. Higher accuracy and lower RMSE are better. We report the mean +/- sem over five seeds. Results indicated with $*$ are significantly worse in performance than bPC determined using an independent-samples t-test (n=5, $p < 0.05$).

| Model | Acc % | | | RMSE | | |
|---|---|---|---|---|---|---|
| | MNIST | Fashion-MNIST | CIFAR-10 | MNIST | Fashion-MNIST | CIFAR-10 |
| bPC | $97.06^{\pm0.08}$ | $88.24^{\pm0.08}$ | $85.06^{\pm0.11}$ | $0.2394^{\pm0.0002}$ | $0.2531^{\pm0.0022}$ | $0.3036^{\pm0.0014}$ |
| hybridPC | $86.15^{\pm0.36*}$ | $80.11^{\pm0.11*}$ | $37.20^{\pm0.43*}$ | $0.2418^{\pm0.0006}$ | $0.2527^{\pm0.0003}$ | $0.3003^{\pm0.0051}$ |
| AE | $98.32^{\pm0.06}$ | $89.49^{\pm0.10}$ | $89.97^{\pm0.12}$ | $0.1576^{\pm0.0005}$ | $0.2164^{\pm0.0003}$ | $0.1599^{\pm0.0002}$ |
| hybridBP | $98.48^{\pm0.03}$ | $89.61^{\pm0.13}$ | $90.11^{\pm0.11}$ | $0.1249^{\pm0.0004}$ | $0.1707^{\pm0.0004}$ | $0.1048^{\pm0.0011}$ |

Table 15: Classification accuracy and class average image generation for bimodal genPC and bPC considered in Section 4.5. Higher accuracy and lower RMSE are better. We report the mean +/- sem over five seeds. Results indicated with $*$ are significantly worse in performance than bPC determined using an independent-samples t-test (n=5, $p < 0.05$).

| Model | Acc % | | RMSE | |
|---|---|---|---|---|
| | MNIST | Fashion-MNIST | MNIST | Fashion-MNIST |
| bPC | $97.80^{\pm0.05}$ | $89.05^{\pm0.10}$ | $0.0506^{\pm0.0007}$ | $0.0431^{\pm0.0004}$ |
| genPC | $88.28^{\pm0.10*}$ | $80.53^{\pm0.14*}$ | $0.1561^{\pm0.0004*}$ | $0.2027^{\pm0.0031*}$ |

## C  OTHER TYPES OF PREDICTIVE CODING MODELS

In this experiment, we train three additional types of predictive coding models on the supervised task described in Section 4.1: (1) PC along arbitrary graphs(Salvatori et al., 2022), (2) bPC with shared weights for discriminative and generative predictions(Qiu et al., 2023), and (3) discPC with activity decay during generation(Sun and Orchard, 2020).

**PC along arbitrary graphs** (agPC) differs from bPC in that each neuron has a single energy function, with predictions computed from all incoming connections. In contrast, bPC uses separate energy terms for bottom-up (discriminative) and top-down (generative) predictions. In this formulation, the energy associated with an agPC layer $x_l$ is given by:

$$E_l = \frac{1}{2} \left\| x_l - f(W x_{l+1} + V x_{l-1} + b) \right\|_2^2,$$

where $W$ and $V$ are top-down and bottom-up weights, respectively. We use the same training algorithm, hyperparameter tuning, and evaluation procedure as bPC, modifying only the energy function. Initialisation is done using bottom-up sweeps (for training and classification evaluation) and top-down forward sweeps (for generation evaluation), matching bPC. We also tested zero and Xavier initialisation. However, this resulted in worse learning performance.

**bPC with shared weights** (shared bPC) uses a single set of weights for both discriminative and generative predictions. Its energy function associated with a layer $x_l$ is given by:

$$E_l = \frac{\alpha_{\text{gen}}}{2} \left\| x_l - f(W_{l+1} x_{l+1}) \right\|_2^2 + \frac{\alpha_{\text{disc}}}{2} \left\| x_l - f(W_l^\top x_{l-1}) \right\|_2^2,$$

where $W_l^\top$ is reused for both directions. This model omits bias terms and relies on non-local computations. We follow the same training, tuning, and evaluation protocol as bPC.

**discPC with activity decay** (decay discPC) extends standard discPC by adding an activity decay term during generation. We use the same training and hyperparameter tuning protocol as discPC. Additionally, we tune an activity decay rate from a log-uniform distribution over the range $[10^{-5}, 1]$. During generative evaluation, we increase the number of inference steps to 10,000 due to slower convergence and we initialise neurons to zero activity before generation for consistency with Sun and Orchard (2020).

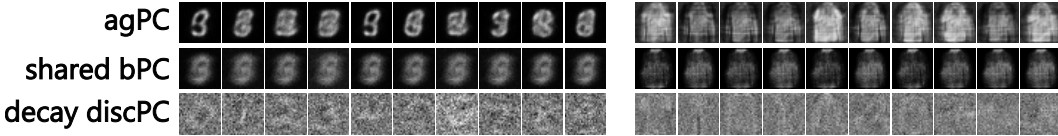

Figure 9: MNIST (left) and Fashion-MNIST (right) class average image generation for agPC, shared bPC and decay discPC.

Table 16 reports classification accuracy and reconstruction RMSE for these models. Figure 9 illustrates the images generated by each model for each MNIST and Fashion-MNIST class. None of the models match bPC in both classification and generative performance.

Table 16: Classification accuracy and class average image generation for agPC, shared bPC and decay discPC on the supervised training task of Section 4.1. Higher accuracy and lower RMSE are better. We report the mean +/- sem over five seeds.

| Model | Acc % | | RMSE | |
|---|---|---|---|---|
| | MNIST | Fashion-MNIST | MNIST | Fashion-MNIST |
| arbitrary graph PC | $71.47^{\pm0.27}$ | $62.50^{\pm6.05}$ | $0.1685^{\pm0.0299}$ | $0.2426^{\pm0.0077}$ |
| shared-bPC | $97.39^{\pm0.08}$ | $87.99^{\pm0.04}$ | $0.1380^{\pm0.0042}$ | $0.2560^{\pm0.0093}$ |
| decay discPC | - | - | $0.4047^{\pm0.0005}$ | $0.3433^{\pm0.0002}$ |

## D  BALANCING DISCRIMINATIVE AND GENERATIVE ENERGIES IN BPC

In this experiment, we investigate how the relative weighting of the discriminative and generative energy terms ($\alpha_{disc}/\alpha_{gen}$) affects bPC's learning in the supervised and unsupervised settings described in Sections 4.1 and 4.2. For each setting, we perform hyperparameter tuning of the bPC model while constraining the ratio $\alpha_{disc}/\alpha_{gen}$, and report the test performance of the model with the optimal parameters. We use a grid search over the following hyperparameters: the activity learning rate [0.01, 0.003, 0.001], the parameter learning rate [0.001, 0.0003, 0.0001], and the weight decay [0., 0.0001, 0.0003, 0.001]. Other settings, including a GeLU activation function and an activity momentum of 0, are constant. All remaining parameters, such as the number of epochs, batch size, and number of inference steps during training and evaluation, are kept consistent with the experiments described in Sections 4.1 and 4.2.

In the supervised case, generation RMSE remains stable across ratios of $\alpha_{disc}/\alpha_{gen}$, but classification accuracy declines as the discriminative energy is down-scaled. We suspect this is due to a mismatch in energy magnitudes. A typical discPC model has an energy of around 0.1 for test data samples after training, while genPC has an energy of approximately 50. Thus, the generative energy dominates when $\alpha_{disc}$ is not significantly larger than $\alpha_{gen}$, leading to a poor classification like genPC models. In contrast, bPC's unsupervised learning performance is relatively robust across a wide range of ratios, but degrades sharply when the discriminative energy is too large. We also observed increased training instability when $\alpha_{disc} \gg \alpha_{gen}$.

These results highlight the importance of appropriately scaling the energy terms based on the task. Hybrid tasks such as those in Section 4.3 require tailored weighting across the models to ensure effective learning. Future work could make the energy scaling a learnable parameter. This change could make bPC learn optimal scalings autonomously.

## E  EFFECT OF MAX-POOLING ON GENERATIVE ARTIFACTS

In this experiment, we modify the VGG-5 architecture of the models from Section 4.3 by removing max-pooling layers and increasing the convolutional strides from one to two. We also increase the

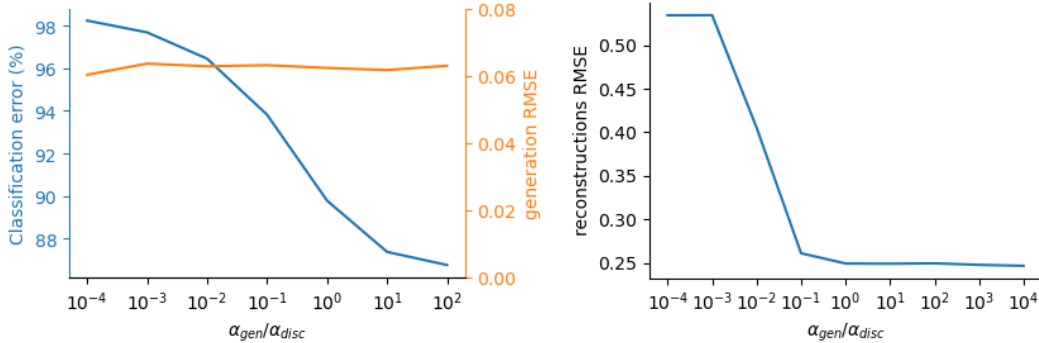

Figure 10: bPC's supervised (left) and unsupervised (right) learning performance on MNIST depending on the relative weighting of its discriminative and generative energy term.

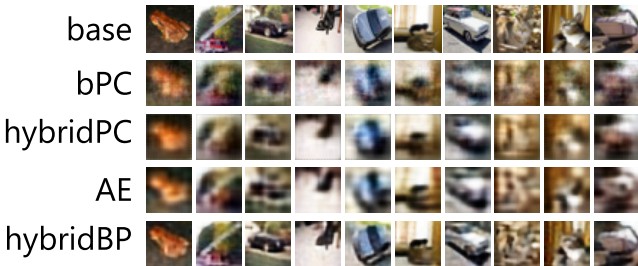

Figure 11: CIFAR-10 image reconstruction from latent representation and labels for models of Section 4.3 without max-pooling.

inference time during training to 48 inference steps. All other aspects of the experiment remain unchanged.

As shown in Figure 11, this change eliminates the checkerboard artefacts in the reconstructed images and improves reconstruction quality for both bPC and hybridPC. This improvement is reflected in the lower RMSE values reported in Table 17. However, removing max-pooling reduces classification accuracy across all models, with bPC experiencing a drop of approximately 25%. Despite this, the overall trends remain consistent: bPC show a substantially better discriminative performance than hybridPC for comparable reconstruction RMSE.

Table 17: Classification accuracy and image reconstruction RMSE from latent representations for CIFAR-10 models considered in Section 4.3 without max-pooling. Higher accuracy and lower RMSE are better. We report the mean +/- sem over five seeds.

| Model | Acc % | RMSE |
|---|---|---|
| bPC | $61.67^{\pm 0.29}$ | $0.2178^{\pm 0.0047}$ |
| hybridPC | $37.36^{\pm 0.18}$ | $0.2196^{\pm 0.0010}$ |
| AE | $84.58^{\pm 0.11}$ | $0.1777^{\pm 0.0005}$ |
| hybridBP | $85.79^{\pm 0.26}$ | $0.1095^{\pm 0.0018}$ |

## F   SAMPLE GENERATION OF COMBINED DISCBP + GENBP MODELS

In this experiment, we repeat the combined model image generation procedure from Section 4.4 using discBP and genBP. The two models share an input layer, which is iteratively updated to minimise their energy functions until the energy falls below the 10th percentile of each model's energy distribution on the test set. We generate samples using the same procedure as described previously

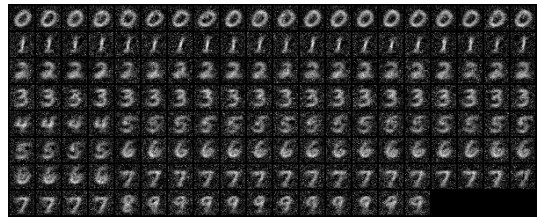

Figure 12: Samples generated for the combined discBP + genBP model.

for the discPC + genPC model. We compute the FID and Inception scores to quantify the generation quality.

Figure 12 displays the generated samples for different classes. While the images resemble the class averages, they also exhibit noise across pixels and less visual diversity than bPC samples. As shown in Table 18, the backpropagation-based combined model yields higher FID and lower Inception scores, indicating that bPC generates samples more consistent with the MNIST distribution than the backpropagation-based approach.

Table 18: Fréchet inception distance and inception score for samples generated for the combined discBP + genBP model compared to bPC and discPC + genPC. Results are given for mean +/- sem for three seeds.

| Model | Inception | FID |
|---|---|---|
| bPC | $6.05^{\pm0.17}$ | $44.4^{\pm2.2}$ |
| discPC + genPC | $3.62^{\pm0.03}$ | $140.5^{\pm2.1}$ |
| discBP + genBP | $5.00^{\pm0.02}$ | $99.1^{\pm3.5}$ |

## G  PARAMETER COUNT MATCHED EXPERIMENTS

In this section, we repeat both the supervised and unsupervised experiments for genPC and discPC, but with parameter counts increased to match bPC. Initially, we maintained identical neural layer dimensions across all models, causing genPC and discPC to have approximately half the number of trained parameters (weight matrices and biases) compared to bPC and hybridPC.

Here, we replicate the MNIST and Fashion-MNIST experiments described in the main paper, adjusting genPC and discPC to match the parameter count of bPC. Specifically, for supervised experiments, the hidden layers of genPC and discPC are expanded to 437 neurons, compared to 256 in bPC. For the unsupervised experiments, genPC layers have 439 neurons. These neuron counts were determined by finding the hidden layer sizes (rounded to the nearest integer) that yield the same parameter count as bPC.

The results, presented in Tables 19 and 20, show that the previous findings remain consistent when parameter counts are matched. GenPC continues to exhibit lower discriminative performance compared to discPC and bPC, as demonstrated by its poorer classification accuracy. Additionally, genPC's unsupervised learning performance remains inferior to that of bPC and hybridPC. DiscPC, similarly, maintains poor generative performance in supervised tasks.

Overall, these experiments confirm that the main paper's conclusion remains valid: bPC effectively integrates the strengths of both genPC and discPC, even when genPC and discPC are scaled to match bPC's parameter count.

## H  RELATIONSHIP BETWEEN HYBRIDPC AND BPC

In this work, we benchmark our bPC model primarily against hybrid predictive coding (Tscshantz et al., 2023). hybridPC is the only plausible PC model that incorporates both slow, iterative inference and fast, feedforward inference, to date. In HybridPC, fast inference is enabled by a bottom-up

Table 19: Classification accuracy and class average image generation for models considered in Section 4.1. All models have approximately the same number of trained parameters. Higher accuracy and lower RMSE are better. We report the mean +/- sem over five seeds.

| | Acc % | | RMSE | |
|---|---|---|---|---|
| **Model** | **MNIST** | **Fashion-MNIST** | **MNIST** | **Fashion-MNIST** |
| bPC | $98.10^{\pm 0.05}$ | $89.24^{\pm 0.12}$ | $0.0581^{\pm 0.0004}$ | $0.0415^{\pm 0.0005}$ |
| hybridPC | $86.22^{\pm 0.15}$ | $80.34^{\pm 0.11}$ | $0.0612^{\pm 0.0003}$ | $0.0480^{\pm 0.0016}$ |
| genPC | $83.09^{\pm 0.80}$ | $78.28^{\pm 0.23}$ | $0.0196^{\pm 0.0002}$ | $0.0142^{\pm 0.0002}$ |
| discPC | $98.68^{\pm 0.04}$ | $89.30^{\pm 0.26}$ | $0.4085^{\pm 0.0040}$ | $0.3209^{\pm 0.0049}$ |
| BP | $98.66^{\pm 0.06}$ | $89.84^{\pm 0.08}$ | $0.0197^{\pm 0.0001}$ | $0.0133^{\pm 0.0001}$ |

Table 20: Image reconstruction RMSE from latent representations for models considered in Section 4.2. All models have approximately the same number of trained parameters. Lower RMSE is better. We report the mean +/- sem over five seeds.

| **Model** | **MNIST** | **Fashion-MNIST** |
|---|---|---|
| bPC | $0.2320^{\pm 0.0010}$ | $0.2497^{\pm 0.0004}$ |
| hybridPC | $0.2401^{\pm 0.0012}$ | $0.2508^{\pm 0.0007}$ |
| genPC | $0.2453^{\pm 0.0007}$ | $0.2861^{\pm 0.0016}$ |

network added to the genPC network architecture, which serves solely to initialise neural activities and does not impact their dynamics.

HybridPC was originally defined with two separate objective functions. One of these is the genPC energy function. The other is a loss for the bottom-up initialisation parameters. However, the learning process of hybridPC can also be expressed as a single unified loss function. This formulation is analogous to the energy function in bPC. The neural dynamics and weight updates in hybridPC minimise the following energy function:

$$E_{hybrid}(x, W, V) = \sum_{l=1}^{L-1} \frac{1}{2}\|x_l - W_{l+1}f(x_{l+1})\|_2^2 + \sum_{l=2}^{L} \frac{1}{2}\|\text{sg}(x_l) - V_{l-1}f(\text{sg}(x_{l-1}))\|_2^2 \quad (9)$$

Unlike in bPC, a stop gradient sg($\cdot$) operation is applied. This change ensures that only the top-down generative predictions drive the neural dynamics. HybridPC can perform both supervised and unsupervised learning. However, the supervised learning performance of hybridPC is much poorer than that of discPC. This is due to the stop gradient operation in the bottom-up stream of this model that prevents it from learning a discriminative model that maps well from $x_1$ to $x_L$ (Tscshantz et al., 2023). , which caps its supervised learning performance to that of genPC (Tscshantz et al., 2023).

In bPC, bottom-up predictions to contribute directly to the inference process, ensuring that the learned neural activity patterns incorporate discriminative signals from the bottom-up pathway, thereby significantly enhancing discriminative performance while retaining strong generative capabilities.

With this reformulation of hybridPC, and given the similarity in objective functions, the bottom-up weight updates in bPC can be interpreted as learning an inversion of the activity updates. This inversion provides a mechanism for fast initialisation in bPC, implementing amortised inference.

## I  SCALING BPC TO DEEPER MODELS

To investigate whether bPC scales effectively to deeper models and more complex datasets, we trained discPC, discBP, and bPC using the following architecture-dataset combinations: VGG-5 on CIFAR-10, VGG-9 on CIFAR-100, and VGG-16 on Tiny-ImageNet.

After training, we evaluated their discriminative performance by measuring classification accuracy. Additionally, we assessed generative capabilities by evaluating classification accuracy when 30% and 50% of the input pixels were missing, following the methodology outlined in Section 4.5.

**Data Preparation**  We used the CIFAR-10 and CIFAR-100 datasets (Krizhevsky, 2009b). For discPC and discBP, images were rescaled to the range [0, 1] and normalized using the mean and standard deviation shown in Table 21, consistent with Pinchetti et al. (2025b). For bPC, images were rescaled to the range [-1,1] to align with the effective output range of the tanh activation function, which served as the output activation for the top-down predictions across all bPC models.

Table 21: Data normalization

|  | **Mean ($\mu$)** | **Std ($\sigma$)** |
|---|---|---|
| CIFAR-10 | [0.4914, 0.4822, 0.4465] | [0.2023, 0.1994, 0.2010] |
| CIFAR-100 | [0.5071, 0.4867, 0.4408] | [0.2675, 0.2565, 0.2761] |
| Tiny-ImageNET | [0.485, 0.456, 0.406] | [0.229, 0.224, 0.225] |

**VGG Architectures**  We utilized deep convolutional neural network architectures from the VGG family (Simonyan and Zisserman, 2014). Table 22 summarises the specific architectures for VGG-5, VGG-9, and VGG-16. Following convolutional layers, a single linear layer was used for classification. We used the GeLU activation function. Batch normalisation was also used in the VGG-9 and VGG-16 models after each convolutional layer to stabilise training.

In the bPC models, the top-down architecture mirrored the discriminative (bottom-up) layers. Each convolutional layer was paired with a transposed convolution sharing identical parameters. However, when convolutional layers were immediately followed by max-pooling operations, the corresponding transposed convolutional layers used a stride of two and a padding of one (instead of zero) to compensate for the change in channel width and height introduced by max pooling. No batch normalisation was used in the top-down predictions.

Table 22: Detailed architectures of VGG models. The locations of the pooling layers correspond to the indices of the convolutional layers after which the max-pooling operations are applied.

|  | **VGG-5** | **VGG-9** | **VGG-16** |
|---|---|---|---|
| Channel Sizes | [128, 256, 512, 512] | [128, 128, 256, 256, 512, 512, 512, 512] | [64, 64, 128, 128, 256, 256, 256, 512, 512, 512, 512, 512, 512] |
| Kernel Sizes | [3, 3, 3, 3] | [3, 3, 3, 3, 3, 3, 3, 3] | [3, 3, 3, 3, 3, 3, 3, 3, 3, 3, 3, 3, 3] |
| Strides | [1, 1, 1, 1] | [1, 1, 1, 1, 1, 1, 1, 1] | [1, 1, 1, 1, 1, 1, 1, 1, 1, 1, 1, 1, 1] |
| Paddings | [1, 1, 1, 1] | [1, 1, 1, 1, 1, 1, 1, 1] | [1, 1, 1, 1, 1, 1, 1, 1, 1, 1, 1, 1, 1] |
| Pool location | [0, 1, 2, 3] | [0, 2, 4, 6] | [1,4, 7,10,13] |
| Pool window | $2 \times 2$ | $2 \times 2$ | $2 \times 2$ |
| Pool stride | 2 | 2 | 2 |

**Learning Rate Schedule**  The learning rate schedule was structured as follows:

1. During the initial 10% of training, the learning rate linearly increased from $w\_lr$ to $1.1 \times w\_lr$.

2. Subsequently, a cosine decay schedule reduced the learning rate smoothly to $0.1 \times w\_lr$ over the remaining epochs.

where refers to the tuned weight learning rate.

**Simulating PC using Error optimisation**  To efficiently simulate the predictive coding models, we employed error optimisation as described in Goemaere et al. (2025). This approach prevents exponential energy decay in predictive coding models and enables the training of larger architectures. discPC directly follows the formulation introduced in the paper. bPC can likewise be expressed using an error reparametrisation: this is done by rewriting the inference energy function of its discriminative component in the same way as for discPC, while leaving the top-down generative prediction error loss unchanged, i.e., computed directly from the layer activities. We validated this reformulation of bPC by confirming that its iterative inference converges to the same equilibrium point as the neural dynamics of bPC described in the main paper, but does so more quickly when using error reparametrisation.

**Model Hyperparameters**  The hyperparameters of discPC and discBP for the VGG5 and VGG9 model were adopted from Goemaere et al. (2025), where they were tuned using Hyperband Bayesian

optimization (via the Weights & Biases platform) across the combinations listed in Table 23. For the VGG16 models, we repeated the tuning summarised in 23.

For bPC models, no additional hyperparameter tuning was performed. Instead, we directly applied the optimal hyperparameters obtained for discPC, while setting bPC's $\alpha_{disc} = 1$, and $\alpha_{gen} = 1e^{-5}$ for VGG-5, and $1e^{-8}$ for VGG-9 and VGG-16.

Table 23: Summary of hyperparameter search space from Goemaere et al. (2025).

| Method | Tuned hyperparameter range | Optimizer | Optim steps (T) | Epochs (sweep/final) |
|---|---|---|---|---|
| discPC | e_lr: fixed at 0.001
e_momentum: fixed at 0.0
w_lr: log-uniform [1e-5, 1e-2]
w_decay: log-uniform [1e-6, 1e-3] | SGD (error)
Adam (weights) | 5 (all models) | 25/25 |
| discBP | w_lr: log-uniform [1e-5, 1e-2]
w_decay: log-uniform [1e-6, 1e-3] | Adam (weights) | NA | 25/25 |

**Glossary:** w_lr: base weight learning rate (see learning rate schedule below), w_decay: weight decay, {e,s}_lr: error / state learning rate, {e,s}_momentum: error / state momentum, T: nr. of optimization steps

**Evaluation**  We evaluated the models under three conditions: with 0%, 30%, and 50% of input pixels set to zero across all image channels. Missing pixels were selected uniformly at random, independent of their spatial locations.

For discPC and discBP models, we obtained classification accuracy directly from a single feedforward pass, as these models inherently yield zero reconstruction loss for any given input.

For bPC models, missing pixel values in $x_1$ were initialised to zero and left unclamped, allowing the network to iteratively infer their values. While 600,000 inference steps were previously used to guarantee convergence for MNIST-trained MLPs, this approach is computationally infeasible for our larger models. To accelerate convergence, we adopted a two-stage inference process. First, we performed 1,000 warm-up inference steps with $\alpha_{disc} = 0$, facilitating faster completion of missing pixel values. Subsequently, we restored $\alpha_{disc} = 1$ and carried out an additional 2,000 inference steps to determine final classification accuracy.

Besides measuring classification accuracy, we qualitatively assessed the ability of bPC models to reconstruct missing inputs by examining post-inference neuron activity in $x_1$.

**Results**  Table 24 shows that bPC achieves classification accuracy comparable to discPC and discBP when full images are presented, even in deeper VGG architectures.

Tables 25 and 26 demonstrate that bPC significantly outperforms discPC and discBP on images with missing inputs. This improvement arises from bPC's iterative inference, which fills in missing information. Figure 13 illustrates this effect: after inference, the activity of the input layer $x_1$ reveals that neurons lacking sensory input have been updated to predict the missing values. Once the missing information is inferred, bPC classifies images accurately. This result holds for both 30% and 50% missing inputs, highlighting bPC's robustness to incomplete data.

Overall, even in larger models, bPC effectively balances discriminative and generative performance by jointly minimising bottom-up and top-down prediction errors.

| Model | bPC | discPC | discBP |
|---|---|---|---|
| VGG5/CIFAR10 | $87.3^{\pm0.3}/98.9^{\pm0.1}$ | $87.6^{\pm0.2}/98.8^{\pm0.1}$ | $87.8^{\pm0.2}/98.8^{\pm0.1}$ |
| VGG9/CIFAR100 | $61.2^{\pm0.5} / 84.9^{\pm0.3}$ | $61.3^{\pm0.5} / 84.6^{\pm0.2}$ | $64.4^{\pm0.1} / 81.9^{\pm0.2}$ |
| VGG16/Tiny-ImageNet | $50.3^{\pm0.4}/73.6^{\pm0.2}$ | $50.0^{\pm0.5}/72.6^{\pm0.2}$ | $47.5^{\pm0.6} / 70.8^{\pm0.3}$ |

Table 24: Classification accuracy of bPC, discPC and discBP when whole images are presented.

| Model | bPC | discPC | discBP |
|---|---|---|---|
| VGG5/CIFAR10 | $\mathbf{80.8}^{\pm 0.3}$ / $\mathbf{97.2}^{\pm 0.1}$ | $18.4^{\pm 1.1}$ / $67.8^{\pm 1.5}$ | $16.9^{\pm 0.4}$ / $63.8^{\pm 1.8}$ |
| VGG9/CIFAR100 | $\mathbf{43.9}^{\pm 1.3}$ / $\mathbf{64.4}^{\pm 1.1}$ | $4.9^{\pm 0.3}$ / $18.1^{\pm 0.7}$ | $4.5^{\pm 0.2}$ / $12.8^{\pm 0.9}$ |
| VGG16/Tiny-ImageNet | $\mathbf{39.2}^{\pm 0.8}$/$\mathbf{55.3}^{\pm 0.5}$ | $18.6^{\pm 0.8}$/$37.4^{\pm 0.9}$ | $17.3^{\pm 0.2}$ / $36.5^{\pm 0.2}$ |

Table 25: Classification accuracy of bPC, discPC and discBP when 30% of presented images are missing. Best result shown in bold.

| Model | bPC | discPC | discBP |
|---|---|---|---|
| VGG5/CIFAR10 | $\mathbf{73.4}^{\pm 0.5}$ / $\mathbf{95.6}^{\pm 0.1}$ | $12.3^{\pm 0.3}$ / $56.4^{\pm 1.0}$ | $12.2^{\pm 0.2}$ / $54.9^{\pm 1.7}$ |
| VGG9/CIFAR100 | $\mathbf{29.4}^{\pm 0.9}$ / $\mathbf{48.0}^{\pm 0.8}$ | $2.5^{\pm 0.3}$ / $10.4^{\pm 0.5}$ | $2.3^{\pm 0.1}$ / $8.2^{\pm 0.2}$ |
| VGG16/Tiny-ImageNet | $\mathbf{27.8}^{\pm 0.5}$ / $\mathbf{41.7}^{\pm 1.1}$ | $7.3^{\pm 0.3}$/$19.3^{\pm 0.2}$ | $7.1^{\pm 0.1}$ / $19.2^{\pm 0.3}$ |

Table 26: Classification accuracy of bPC, discPC and discBP when 50% of the presented images are missing. Best result shown in bold.

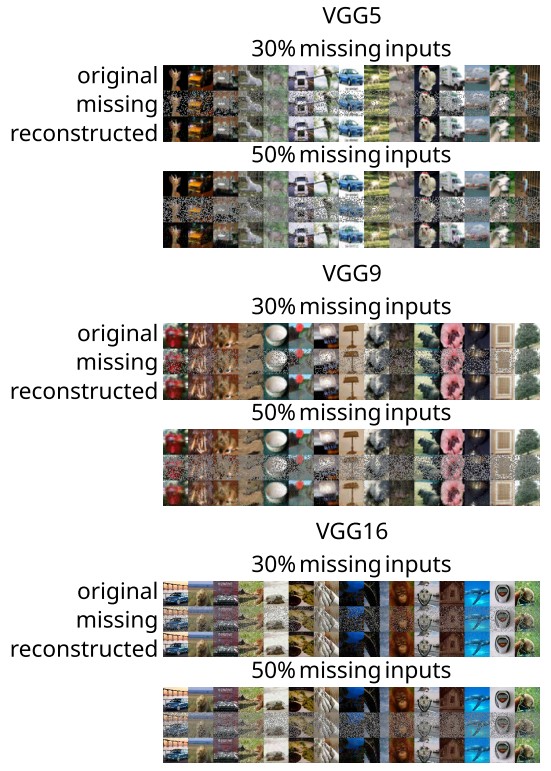

Figure 13: Image reconstruction through iterative inference by bPC.

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
