# OpenReview forum: "Bidirectional Predictive Coding"
_ICLR.cc/2026/Conference — ICLR 2026 Poster_

### Official Review · Reviewer_QM9z · 2025-10-24

**Soundness:** 3
**Presentation:** 3
**Contribution:** 3
**Rating:** 8
**Confidence:** 4

**Summary:**

This paper proposes a hierarchical neural network architecture in which the same error-minimization process is used to reconcile both bottom-up and top-down signals. Neural activity is determined using an iterative process of gradient decent to minimize these errors. Weights are also updated to minimize theses errors.

**Strengths:**

The proposed architecture employs only local information for inference and learning.

The proposed architecture is very flexible and can be used for a range of tasks, such as image classification, image generation, supervised and unsupervised learning.

**Weaknesses:**

Fig. 1 is not very helpful in showing the differences between architectures. One issue is that arrows of the same colour seem to mean different things in different panels.

The method is only tested using small, simple, data-sets.

Many inference steps are performed to process each sample. The number of steps seems too large for this to be a viable model of biological visual inference or a method that could have practical, machine learning, applications.

The relatively small differences in performance are difficult to see given the large scales used in the graphs showing the results (Figs. 3, 4, and 5).

**Questions:**

The proposed architecture seems to be very similar to that proposed in Qiu et al. (2023). Apart from the lack of tied weights (mentioned on l.155) are there any other differences? Why not compare the performance of your network to this one?

Could the proposed method scale-up to deal with large data-sets, such as ImageNet?

How was the number of iterations chosen? How sensitive are the results to this choice? Would the number of inference steps need to increase with the size of the architecture?

How do the computational costs of the different methods compare?

Could the proposed method be used for semi-supervised learning?

It is claimed (l.292) that the proposed architecture only requires half as many neurons as an alternative architecture using separate bottom-up and top-down networks. Is this claim ignoring the error-neurons? If so, why?

What do the error bars on the results figures represent? Are the differences in performance of the compared methods statistically significant?

---

> ### Author Response · Authors · 2025-11-21
>
> We thank the reviewer for the valuable feedback. We have addressed all the comments, and we believe the revisions will substantially enhance the clarity of our results.
>
> **Comments:**
>
> **- Fig. 1 is not very helpful in showing the differences between architectures.**
>
> We thank the reviewer for this helpful observation.  To make the difference between predictive coding and backpropagation models more visually apparent, we have updated Figure 1 to visualise the flow of error signals. This revision should make it clear that PC models only rely on local errors while BP models rely on global error signals.
>
> **- Why not compare the performance of your network to Qiu et al. (2023)?**
>
> Thank you for raising this point. Qiu et al. (2023) perform poorly in practice because the shared set of weights must simultaneously optimise two competing objectives. We had included experiments comparing bPC with Qiu et al. in SM C. To make these results more visible in the manuscript, we have added a clear mention in l256-259.
>
>
> **- Could the proposed method scale-up to deal with large data-sets, such as ImageNet? The number of steps seems too large for this to be a viable model of biological visual inference or a method that could have machine learning applications. How was the number of iterations chosen? How sensitive are the results to this choice? Would the number of inference steps need to increase with the size of the architecture?**
>
> Thank you for these questions. We expect bPC to scale well to deeper architectures and larger datasets. Preliminary results showed that bPC scales to deep models (up to VGG16) and larger datasets such as Tiny-ImageNet. Originally included in SM I, have now moved these to section 4.6.
>
> Regarding biological plausibility, iterative inference is expected to be viable in the brain: sensory inputs are highly temporally correlated, so only a few iterations are needed between inputs. If an unexpected input is presented, the brain could rapidly process it through bPC’s bottom-up activity initialisation before refining it iteratively.
>
> For large-scale datasets such as ImageNet, the primary limitation is the computational cost of iterative inference. In our experiments, we used the minimum number of steps needed for a performance comparable to backpropagation. Increasing the number of steps improves bPC's performance, and deeper models require more inference steps.
>
> To address scalability challenges, we identify three promising directions:
>
> 1) Improved simulation techniques: Recent advances (Goemaere et al., 2025) have substantially reduced inference costs, making large-scale PC experiments increasingly feasible. We have adopted these techniques to demonstrate that bPC scales to deeper architectures.
>
> 2) Optimised PC libraries: Predictive coding can, in principle, be fully parallelised, unlike backpropagation, which requires sequential passes across layers. However, PC libraries do not support full parallelisation yet.
>
> 3) Hardware specialisation: Existing digital hardware is not optimised for iterative inference loops. Neuromorphic or analog hardware that naturally supports such dynamics could significantly improve speed and energy efficiency.
>
>
> **- Could the proposed method be used for semi-supervised learning?**
>
> Thank you for the comment. Semi-supervised learning is a promising direction for bPC, as the model naturally supports both supervised and unsupervised learning within a single biologically plausible architecture. We are considering two approaches:
>
> 1) Temporarily removing bPC's classifier head during unsupervised updates so the model behaves as in our unsupervised experiments (Section 3.2). With labelled samples, periodically reintroducing the classifier and its top-down label signal to align representations.
>
> 2) Keeping the classifier head but fixing its predicted labels during unlabeled inference, effectively treating them as pseudo-labels. This maintains consistent dynamics across phases.
>
> **- It is claimed that the proposed architecture only requires half as many neurons. Is this claim ignoring the error-neurons?**
>
> Thank you for raising this question. Our claim omits error neurons because some PC implementations treat error computation as a dendritic process (Mikulasch et al., 2023), so additional error terms increase dendritic activity rather than neuron count. Other formulations model error units explicitly.
>
> To make our statement less tied to the neural implementation, we have revised it in l301-305.
>
>
> **- What do the error bars on the results figures represent? Are the differences in performance statistically significant?**
>
> Thank you for the comment. The error bars represent the standard error of the mean across five seeds. We have clarified this in the captions. The performance differences between the compared methods are statistically significant, determined by independent-samples t-tests. We have included the results of these tests in Appendix B.

---

> > ### Comment · Reviewer_QM9z · 2025-11-26
> >
> > Having read the detailed rebuttal to myself and the other reviewers, I remain confident that this paper deserves to be accepted, and so will keep my score unchanged. I disagree with Reviewer Pgr9, in that I think it is possible to propose interesting (biologically-inspired) innovations that don't (immediately) need to produce state-of-the-art results. It is possible that interesting and important ideas may take decades of work to reach good performance on benchmarks, just as it took CNNs decades to achieve good performance.

---

### Official Review · Reviewer_UixL · 2025-10-27

**Soundness:** 4
**Presentation:** 3
**Contribution:** 4
**Rating:** 10
**Confidence:** 5

**Summary:**

The brain is able to do both generative and discriminative tasks, however the current most influential model for how the brain does that, predictive coding, has great trouble doing both in the same network. The authors present a surprisingly simple bidirectional version of the classical Rao&Ballard predictive coding model and show how that then a single model is able to solve both generative and discriminative tasks.

**Strengths:**

This is a great paper. Conceptually simple, though with many smaller innovations that are only mentioned in passing, the presented work marks a great step towards highly functional predictive coding networks, fixing previous issues with capabilities and, importantly, scaling.
The work is exceptionally thorough, well designed and -mostly- very clear (see below).
Noteworthy is also the work in the appendix where the models are scaled up to large networks and many-class classification benchmarks. This solves another big problem with predictive coding networks, that is, the previous work simply does not scale (we, and others, tried). For example, the approach by Song et al mentioned fails for cifar-100 and also for deeper networks. The approach presented here does scale, and this should be mentioned more clearly in the final version of the paper.

**Weaknesses:**

Minor issues:
- the issue of symmetric/shared bottom-up/top-down connections (l154) is a bit exaggerated, as it is easy to show that as long as identical local learning signals can be applied at either end, weight decay can result in symmetrical weights. (Tim Lillicrap wrote on this).
- section 4.5 lacks clarity, in particular on what exactly is shown in Figure 7. What am I seeing here, what is the paradigm?
- the issue of scaling, which this approach solves to a large extend, is only mentioned in the appendix.

**Questions:**

- beyond fig 4e, what is the influence of the number of inference steps for the various problems, and is inference done sequentially per layer or in parallel?
- can you say something on whether/how the approach would extend to dynamic rather than static stimuli?
- can you comment more on how the networks are initialized for unsupervised learning (l106/l302)?
- the VAE may not be the best self-supervised model for both generative and discriminative tasks within the set framework, that seems to rather be Masked-Autoencoders. Can you comment on this? Ideally compare to MAEs?
- In 4.3, it is noted that learning rarely takes place only in an unsupervised setting. I would argue however that supervised learning is much rarer than RL. Can you comment on whether the approach extends to RL?
- the fact that Max-Pooling has such detrimental results is noteworthy (l 364), and would seem to argue for for example all-convolutional networks. Comment?

---

> ### Author Response · Authors · 2025-11-21
>
> We thank the reviewer for the valuable feedback. We have addressed all the comments, and we believe the revisions enhance the manuscript.
>
>
> **- the issue of symmetric/shared bottom-up/top-down connections (l154) is a bit exaggerated.**
>
> Apologies for not being clear in l154. We meant:
>
> "
> Finally, Qiu et al. (2023) proposed a bidirectional PC model in which the connections between separate layers share the same weights, e.g. the bottom-up weights from layer $l-1$ to $l$ equal the top-down weights from $l$ to $l+1$. It is unlikely that the brain shares synaptic connections between separate layers of processing.
> "
>
> **- section 4.5 lacks clarity, in particular on what exactly is shown in Figure 7.**
>
> We appreciate the reviewer’s comment regarding clarity.
>
> In this section, we consider two settings inspired by the human brain. First, we study learning in an architecture with two separate input streams, mirroring how different sensory modalities are processed. Second, we assess the robustness of bPC under incomplete information, akin to how vision compensates for the retinal blind spot.
>
> To improve clarity of this section, we have revised Section 4.5 by adding an introduction paragraph (l463-466) and including titles in Figure 7.
>
>
> **- the issue of scaling, which this approach solves to a large extend, is only mentioned in the appendix.**
>
> We thank the reviewer for this valuable observation. To better highlight our contribution in this direction, we have moved the results from Appendix I to the Results, see 4.6.
>
> **- beyond fig 4e, what is the influence of the number of inference steps for the various problems, and is inference done sequentially per layer or in parallel?**
>
> We thank the reviewer for this question. Across tasks, increasing the number of inference steps improves learning stability and performance. In our current implementation, inference runs sequentially across layers because existing PC libraries lack parallel support. Enabling parallel inference in these libraries would be a valuable direction for future work.
>
>
> **- Can you say something on whether/how the approach would extend to dynamic rather than static stimuli?**
>
> Extending bPC to dynamic stimuli represents an advancement towards more realistic models of vision. bPC could be extended in two possible ways: (i) using generalised coordinates (Friston, Trends in Cogn. Sci., 2009), or (ii) using temporal predictive coding (Millidge, PLOS CB, 2024). We expect bPC to work well in dynamic settings because this change would bring it close to a PredNet model (Lotter, ICLR, 2017), which is effective on dynamic tasks.
>
> **- can you comment more on how the networks are initialized for unsupervised learning (l106/l302)?**
>
> We thank the reviewer for raising this point. For unsupervised learning, bPC's neural activity is initialised using a bottom-up feedforward sweep. This initialisation substantially reduces the number of inference steps required for good performance by initialising neurons close to their steady state.
>
> For clarity, we have changed the description of bPC's activity initiation in l195-201.
>
>
> **- the VAE may not be the best self-supervised model for both generative and discriminative tasks within the set framework, that seems to rather be Masked-Autoencoders.**
>
> We appreciate the reviewer’s observation. Masked Autoencoders are indeed powerful self-supervised models.
>
> Our baselines use backpropagation-trained autoencoders to isolate the effect of the learning rule itself. The BP and PC models share the same architecture and near-identical training schedules, allowing a clean comparison between PC and backpropagation.
>
> MAEs would complicate this comparison, as their additional inductive biases change architecture and training dynamics, making it harder to attribute performance differences to the learning algorithm alone.
>
> That said, integrating masking objectives into bPC is an interesting direction that could yield richer representations.
>
> **- I would argue that supervised learning is much rarer than RL. Can you comment on whether the approach extends to RL?**
>
>
> Our experiments suggest that bPC can be applied in the same settings where discPC has succeeded previously, and therefore should naturally extend to reinforcement learning. In particular, discPC outperforms backpropagation in some RL tasks (Song et al., Nature Neuroscience, 2025).
>
> **- the fact that Max-Pooling has such detrimental results is noteworthy (l 364), and would seem to argue for example all-convolutional networks.**
>
> We agree with the reviewer. Pooling removes spatial details beneficial for bPC. This motivates architectures with minimal pooling, such as ResNets. Although early predictive coding ResNets faced stability issues (Pinchetti, ICLR 2025), recent work provides stable parameterisations (Innocenti, NeurIPS 2025). This suggests that all-convolutional residual architectures may further improve bPC’s scalability.

---

### Official Review · Reviewer_Pgr9 · 2025-10-29

**Soundness:** 2
**Presentation:** 3
**Contribution:** 1
**Rating:** 2
**Confidence:** 4

**Summary:**

This paper proposes Bidirectional Predictive Coding (bPC), a biologically inspired architecture that aims to integrate discriminative and generative learning within a single predictive coding framework. The authors claim that bPC can simultaneously achieve high classification accuracy and realistic image generation, and present results across supervised, unsupervised, and multimodal settings using datasets such as MNIST, Fashion-MNIST, CIFAR-10, and CIFAR-100.
Although the paper is clearly written and the experiments are extensive, the overall novelty and empirical significance of the work are limited. The methodological design and experimental choices do not convincingly demonstrate contributions beyond existing predictive coding or joint generative–discriminative approaches.

**Strengths:**

1. The idea of integrating bidirectional inference under a unified predictive coding framework is conceptually clear and aligns with prior theoretical literature.
2. The paper reports a broad range of experiments, including classification, image generation, multimodal learning, and occlusion robustness, showing some effort toward comprehensive evaluation.

**Weaknesses:**

1. The central idea of combining discriminative and generative pathways within a shared latent representation is not novel in the context of modern machine learning. Many recent architectures, such as VAVAE[1] or VAR[2], already unify discriminative and generative learning with better theoretical grounding and empirical performance. The proposed bPC model appears to be a minor variation of existing predictive coding formulations (discPC, genPC, hybridPC) with shared weights, rather than a fundamentally new mechanism.
2. The network design is extremely simple. Most experiments are conducted with two hidden layers of 256 neurons or shallow convolutional networks. This small scale makes it difficult to evaluate whether the claimed properties would generalize to larger or more realistic settings.
3. The choice of benchmarks (MNIST, Fashion-MNIST, CIFAR-10, CIFAR-100) is outdated and insufficient to support strong claims of scalability or generality. While such datasets can be acceptable for conceptual neuroscience modeling, they are not appropriate for evaluating new machine learning architectures, especially in a venue like ICLR which emphasizes methodological innovation and empirical rigor in AI.
4. Although biologically inspired models sometimes employ simplified architectures to study neural mechanisms, those works typically analyze real neural data or test neuroscientific hypotheses. This paper does neither. Without such grounding, the justification for using small-scale networks and toy datasets becomes unclear and weakens the relevance of the work to both neuroscience and AI audiences.
5. In Figure 1, subfigures (A) and (B) show no visible difference between discPC and discBP, or between genPC and genBP, making it difficult for readers to understand the structural differences among these variants.
6. In Figure 7(A), the authors mention using one-hot labels as text inputs. However, multimodal setups typically require a meaningful text encoder (e.g., CLIP). Using one-hot vectors does not constitute true multimodal learning, and the symbol shown in the figure (an ear icon) is also misleading in this context.
7. Many mathematical expressions lack punctuation marks at the end of equations.

[1] Reconstruction vs. Generation: Taming Optimization Dilemma in Latent Diffusion Models. CVPR 2025

[2] Visual autoregressive modeling: Scalable image generation via next-scale prediction. NeurIPS 2024

**Questions:**

See Weaknesses.

---

> ### Author Response · Authors · 2025-11-21
>
> We thank the reviewer for the valuable feedback. We have carefully addressed all comments and believe the revisions significantly improve the conceptual motivation and neuroscientific relevance of our proposed model.
>
> **Comments:**
>
> **- Although biologically inspired models sometimes employ simplified architectures to study neural mechanisms, those works typically analyze real neural data or test neuroscientific hypotheses. This paper does neither.**
>
> Thank you for these thoughtful comments. We aim to offer a conceptual contribution to neuroscience: we examine how the brain might learn using only local signals and how such mechanisms could support both generative and discriminative inference within a single circuit. The work is intended as biologically grounded theory rather than a new machine-learning architecture.
>
> Our study focuses on two core questions:
>
> 1. Can learning be fully local?
> Backpropagation requires non-local information and is therefore biologically implausible. Our model instead uses local computations and Hebbian plasticity (Section 3.1, Fig. 2).
>
> 2. Can one circuit support both inference modes?
> Experimental evidence indicates that cortex carries out bottom-up and top-down inference simultaneously. bPC provides a mechanistic account of this, demonstrating that a single network can perform both forms of inference across a range of tasks (Sections 4.1–4.4).
>
> Recent findings also strengthen the biological grounding of bPC: dual error-neuron populations observed in cortex (Jordan and Keller, 2020) mirror the two error types in our model. Whereas some studies attribute them to physiological constraints (Keller and Mrsic-Flogel, 2018), in bPC they arise naturally from computational principles.
>
> To make the scope of the work explicit, we have revised the Introduction to emphasise that this is a neuroscience study of biologically plausible learning (l59–64).
>
>
> **- The choice of benchmarks is outdated. While such datasets can be acceptable for conceptual neuroscience modeling, they are not appropriate for evaluating new machine learning architectures.**
>
> Thank you for the thoughtful feedback. Regarding the simplicity of the networks and datasets: all recent predictive coding models published at ICLR, including Pinchetti et al., ICLR 2025 (Spotlight), Salvatori et al., ICLR 2024 (Poster), and Millidge et al., ICLR 2023 (Poster), have used similar architectures and benchmarks. These tasks are standard for conceptual neuroscience PC modeling.
>
> Nonetheless, we had included additional experiments on scaling (model depth up to VGG-16 and dataset complexity with tiny-ImageNet). These results were originally included in Appendix I. We have now moved these results into the new Results section 4.6.
>
> **- The central idea of combining discriminative and generative pathways within a latent representation is not novel in the context of modern machine learning.**
>
> We agree that combining discriminative and generative pathways is not new in machine learning. However, our contribution lies in achieving this integration within a biologically plausible framework. This biological grounding differentiates bPC from backpropagation-trained architectures such as VAVAE or VAR, which do not adhere to neurobiological locality constraints.
>
> To clarify this point, we have added a paragraph to the Background section, see l173-180.
>
>
> **- The proposed bPC model appears to be a minor variation of existing predictive coding formulations.**
>
> We thank the reviewer for pointing out the need to clarify this point. The key contribution of bPC is the unification of generative and discriminative predictive coding within a single energy function and a biologically plausible circuit. This allows the same circuit to perform both supervised and unsupervised learning, thereby more closely mirroring the integrative nature of cortical computation.
>
> We have emphasised this distinction by revising the description of bPC in the manuscript (see l184-188).
>
> **- In Figure 1, subfigures (A) and (B) show no visible difference making it difficult to understand the differences among variants.**
>
> Thank you for this helpful comment. We have revised Figure 1 to highlight error-signal flow, making the contrast clear: PC uses local signals, whereas BP depends on global ones.
>
>
> **- In Figure 7(A), the authors mention using one-hot labels as text inputs. Using one-hot vectors does not constitute true multimodal learning.**
>
> Thank you for the helpful clarification. In our manuscript, “multimodal” was used in the neuroscientific sense to refer to a model architecture with distinct sensory input modalities. In this simplified setting, the one-hot label stream represents a high-level cortical “auditory” or categorical signal rather than linguistic text embeddings.
>
> To avoid misinterpretation, we now describe this setup as a bimodal architecture rather than multimodal learning. We also added an introduction paragraph to Section 4.5 clarifying this (l463–466).

---

> > ### Comment · Reviewer_Pgr9 · 2025-11-24
> >
> > I appreciate the authors’ detailed rebuttal. Several of my earlier questions, particularly those regarding the biological motivation of the work, have been addressed, and the manuscript is now clearer in scope. However, my main concerns remain only partially resolved:
> > 1. Even though the authors present the method in a biologically motivated context, the core idea is already common in many modern deep learning architectures that combine generative and discriminative learning within shared representations. As a result, the novelty and algorithmic contribution remains limited.
> > 2. The issue of scalability is still a major concern. I understand the authors’ argument that many predictive coding papers use small architectures and datasets. However, my concern is not about matching field norms, but about whether the results offer meaningful value for an AI-focused venue. Even neuroscience-inspired research can be assessed at contemporary machine-learning scales. For instance, Memory Mosaics at Scale[1] (NeurIPS 2025) is biologically motivated yet demonstrates its effectiveness on models up to the level of Llama-8B. Without evaluation on larger models and more challenging datasets, it is still difficult to assess whether the insights here extend beyond small toy settings. The newly added VGG-16 and tiny-ImageNet experiments help, but they are limited and do not fully resolve the scalability question.
> >
> > Taking these points together, I am raising my score from 2 to 4, reflecting that the paper is clearer after rebuttal but still limited in novelty and practical impact for an AI-focused venue.
> >
> > [1] Memory Mosaics at Scale. NeurIPS 2025

---

### Official Review · Reviewer_QuQM · 2025-11-01

**Soundness:** 4
**Presentation:** 3
**Contribution:** 3
**Rating:** 6
**Confidence:** 5

**Summary:**

This paper proposes a bidirectional predictive coding (PC) model. Unlike Rao and Ballard’s formulation, where feedforward signals primarily carry prediction errors and feedback connections carry predictions, this model allows both feedforward and feedback pathways to carry predictions or expectations. The latent activity at each level is jointly constrained by top-down expectations and bottom-up proposals, with the energy function minimizing both discrepancies. This formulation brings predictive coding closer to classical interactive activation models (McClelland and Rumelhart), adaptive resonance theory (Grossberg and Carpenter), and hierarchical Bayesian inference via bottom-up and top-down belief propagation (Lee and Mumford). While the conceptual shift is incremental, the formulation is clean and biologically motivated, and the empirical comparisons establish the model’s competitiveness relative to other predictive coding models, particularly the hybrid predictive coding schemes.

**Strengths:**

1. The bidirectional predictive coding formulation is conceptually simple, elegant, and aligns more closely with classical interactive cortical models. It is worth emphasizing, however, that in Rao and Ballard’s model, the initial feedforward sweep does convey bottom-up evidence (the bottom-up prediction), and only subsequent iterations carry the prediction residues or prediction error signals. Thus, the primary difference is in the steady-state treatment and symmetry of information flow.

2. The inference and learning rules are derived from an energy function, and learning is Hebbian, giving this approach greater biological plausibility than backprop-based deep networks.

3. In contrast to autoencoder architectures, encoding and decoding (recognition and generation) in this model are unified. The same circuitry supports inference, classification, generation, and cross-modal completion without explicit mode-switching.

4. Demonstrating cross-modal pattern completion within a predictive-coding framework is novel and compelling.

5. From a neuroscience perspective, the model avoids assuming separate error and representation neurons; instead, each unit receives both bottom-up and top-down drives and evolves dynamically to reduce error. This yields an elegant single-population interpretation consistent with cortical microcircuit ideas.

**Weaknesses:**

1. The conceptual innovation is modest and closely related to earlier models of bidirectional inference (interactive activation, adaptive resonance, hierarchical Bayesian models). The paper could better situate itself in this lineage.

2. Although the results show benefits over prior predictive-coding variants, scalability to real-world architectures and large-scale deep learning benchmarks remains unclear; current experiments appear focused on moderate-scale or simplified tasks.

3. Historically, biologically motivated predictive coding architectures such as PredNet (Lotter, Kreigman, Cox) achieved strong performance in video prediction. It would be useful to discuss the connection and differences, particularly regarding information flow vs. error flow debates.

**Questions:**

1. How do the authors position this model relative to classical bidirectional inference frameworks such as interactive activation, adaptive resonance, hierarchical Bayesian inference, and encoder-decoder networks with skip pathways (e.g., U-Net), all of which integrate bottom-up and top-down signals during inference?

2. Prior work (e.g., PredNet) argued that whether feedforward activity represents proposals or prediction errors may not significantly change performance in video-prediction tasks. Have you evaluated a variant using pure error propagation feedforward, and how do the results compare?

3. What are the main obstacles to scaling this architecture to large-scale tasks comparable to modern deep nets? Are the challenges computational, architectural, representational, or learning algorithms?

---

> ### Author Response · Authors · 2025-11-21
>
> We thank the reviewer for the valuable feedback. We have addressed all comments carefully and believe the revisions substantially improve the scope of our work.
>
> **Comments:**
>
> **- How do the authors position this model relative to classical bidirectional inference frameworks?**
>
> We appreciate this thoughtful question about the relation between bPC and earlier bidirectional inference frameworks. Although classical bidirectional approaches share the idea of combining top-down and bottom-up signals, bPC differs in both its biological motivation and its unified predictive-coding formulation.
>
> 1. Interactive Activation Models (IAM): IAMs use fixed weights and lateral inhibition to stabilise states. bPC instead relies on Hebbian plasticity and an energy-based objective, enabling a single mechanism for both supervised and unsupervised learning.
>
> 2. Adaptive Resonance Theory (ART): ART employs top-down templates and vigilance signals but lacks fully bidirectional prediction and is usually restricted to shallow category learning. bPC performs hierarchical bidirectional inference across multiple layers.
>
> 3. Hierarchical Bayesian Inference: Traditional predictive-coding and Bayesian models use top-down priors with bottom-up error signals. bPC extends this by jointly optimising generative and discriminative energies, allowing genuinely bidirectional inference and improved learning.
>
> 4. Encoder–Decoder Networks (e.g., U-Net): Such networks combine upward and downward pathways but depend on global backpropagation and maintain separate representational streams. bPC uses shared latent states and local learning rules within a single biologically inspired circuit.
>
> We have updated the background section accordingly, expanding discussion of relevant neuroscience models and adding a paragraph on machine-learning architectures such as U-Net, see lines 158 to 180.
>
> **- Prior work (e.g., PredNet) argued that whether feedforward activity represents proposals or prediction errors may not significantly change performance in video-prediction tasks. Have you evaluated a variant using pure error propagation feedforward, and how do the results compare?**
>
> We thank the reviewer for this insightful comparison. While PredNet and bPC share the goal of modeling bidirectional processing, their underlying assumptions differ.
>
> Our primary aim is to investigate how a biologically plausible circuit can support both fast bottom-up discriminative inference and slower top-down generative refinement.
> Adapting bPC to propagate errors upwards, as in PredNet, would eliminate bPC’s fast bottom-up inference mechanism, as it would no longer retain a direct bottom-up prediction pathway. This would compromise the biological interpretability of our model and likely decrease the model's learning performance for an equal number of inference steps.
>
> As PredNet is very relevant to our work, we have now included PredNet and related bidirectional machine learning models in the Background section, see l176.
>
>
> **- What are the main obstacles to scaling this architecture to large-scale tasks comparable to modern deep nets?**
>
> We appreciate this important question. The main obstacle to scaling predictive coding is computational: unlike backpropagation, it requires iterative inference, which is costly on standard digital hardware.
>
> We see three routes to mitigating this:
>
> 1. Faster simulation methods. Recent work on efficient PC inference (e.g., Goemaere et al., 2025) greatly lowers computational cost. Using these techniques, we show that bPC scales to deeper networks (up to VGG-16) and larger datasets (e.g., Tiny-ImageNet). These findings, originally in Appendix I, now appear in the main Results section (see 4.6).
>
> 2. Better PC libraries. Predictive coding is fully parallelisable in principle, and early implementations already show substantial speedups (e.g., Salvatori, ICLR 2024). Current libraries remain limited, for example, full parallelisation often requires uniform layer widths (Pinchetti, ICLR 2025), but further library optimisation could narrow the gap with backpropagation.
>
> 3. Specialised hardware. Digital hardware is poorly matched to PC’s iterative dynamics. Neuromorphic or analogue systems could execute these loops far more efficiently, enabling scalable and energy-efficient PC models.

---

### Meta-Review · Area_Chair_ZdSb · 2025-12-15

**Summary:**

This paper proposes bidirectional predictive coding (bPC), which unifies generative and discriminative inference within a single biologically plausible architecture using local learning rules.

The reviewers are split in their assessment.
* Two reviewers (UixL: 10/10, QM9z: 8/10) are strongly positive, commending the conceptual clarity, biological plausibility, and demonstration that bPC solves previous scaling issues with predictive coding networks.
* One reviewer (QuQM: 6/10) finds the work solid but notes modest conceptual innovation relative to classical bidirectional models.
* One reviewer (Pgr9: 4/10 after rebuttal) has significant concerns about novelty in the ML context and limited empirical evaluation despite biological motivation.

The main points of contention are:
1. whether the biological framing justifies evaluation on small-scale benchmarks without neural data analysis,
1. the degree of novelty given existing bidirectional inference frameworks in both neuroscience and ML, and
3. whether the scaling results (VGG-16, Tiny-ImageNet) adequately demonstrate practical viability.

**Reviewer Concerns:**

**Addressed by rebuttal:**

* Positioning relative to classical bidirectional models (IAM, ART, hierarchical Bayesian inference): Authors added comprehensive discussion in Background section (lines 158-180).
* Clarity of Figure 1 and Section 4.5: Revised to better visualize error signal flow and clarify bimodal architecture setup
* Scaling limitations: Authors moved Appendix I results to main text (Section 4.6), demonstrating scaling to VGG-16 and Tiny-ImageNet
* Biological grounding: Authors clarified this is neuroscience theory work focused on local learning and unified circuits (lines 59-64)
* Statistical significance of results: Added t-test results in Appendix B

**Outstanding concerns:**

* Limited novelty in ML context: While authors clarify biological plausibility distinguishes bPC from models like VAVAE/VAR, Pgr9 maintains that the core idea of combined generative/discriminative learning in shared representations is not novel
* Scalability to truly large-scale tasks: Despite VGG-16/Tiny-ImageNet results, questions remain about scaling to ImageNet-scale or modern deep learning benchmarks. The computational cost of iterative inference remained a limitation/point of discussion
* Lack of neural data analysis: For a biologically-motivated paper, no comparison with actual neural recordings was performed beyond conceptual alignment

**Reviewer Scores:**

* QuQM (6→6): Concerns were partially addressed. The expanded context and scaling results are positive, but the reviewer's core assessment of "modest innovation" likely remains. Score would thus likely stay unchanged.
* Pgr9 (2→4): Explicitly increased score after rebuttal due to improved clarity, but maintains rejection recommendation due to limited novelty and insufficient demonstration of value for an machine learning/AI-focused venue.
* UixL (10→10): Very enthusiastic. Rebuttal addressed minor concerns about clarity.
* QM9z (8→8): Positive assessment maintained after rebuttal. Explicitly stated disagreement with Pgr9's requirement for immediate SOTA results from biologically-inspired work.

---

### Decision · Program_Chairs · 2026-01-26

Accept (Poster)